

# Semantic Description and Complete Computer Characterization of Structural Geological Models

Xianglin Zhan[1], Jiandong Liang[2], Cai Lu[1], Guangmin Hu[2],

[1] School of Information and Communication Engineering, University of Electronic and Science Technology of China, Chengdu, 611731, China
[2] School of Resources and Environment, University of Electronic and Science Technology of China, Chengdu, 611731, China

*Correspondence to*: Jiandong Liang (jd.liang@yahoo.com)

**Abstract.** A structural geological model is an important basis for the understanding of subsurface structures and exploration of mineral resources, especially petroleum reservoirs. In the field of geological modelling, the lack of a well-defined semantic level description and corresponding computer characterization method hinders its application. In this paper, we propose the semantic descriptions for structural geological models in order to facilitate computer based processing of geological semantics. A multi-level heterogeneous network is proposed to characterize the semantic description for this purpose. The semantic description of a structural geological model gives a complete description of structural units (called semantic entities) of structural models. Basic semantic entities include points, lines, interfaces, bodies, formations and advanced semantic entities include stratified structures/massive structures, planar structures, linear structures. Semantic relations represent the logical relationships among these semantic entities. The multi-level heterogeneous network contains complete information of structural geological models for both geometry and geology. Hence, it has a one-to-one correspondence with a structural geological model. In particular, we propose a bottom-up and top-down integrating structural modelling method based on semantic descriptions. This approach aims to address defects of the existing structural modelling methods that can only carry out bottom-up modelling. Because the addition of semantic information, it improves the adaptability of structural modelling to complex structures and enhances modelling efficiency.

## 1 Introduction

Three-dimensional structural geological model is an important way to describe subsurface structures. It provides a basic constraint framework for sequence modelling, lithofacies modelling and reservoir description. It is the foundation for making and optimizing exploration and development schemes in oil and gas resource exploration (Alcalde et al., 2017; Bond, 2015; Lemon and Jones, 2003; Nikitin et al., 2018; Tahmasebi and Kamrava, 2018). Houlding (1994) proposed the concept of three-dimensional geological modelling. Lemon and Jones (2003) introduced a simple way to generate solid models from borehole data. Wu et al. (2005) proposed a multi-source data integration and gradually refined 3D modelling method. Frank





et al. (2007) proposed a method of implicitly reconstructing geometric shape from point cloud data. Caumon et al. (2009) proposed a general process for reconstructing structural models consisting of faults and horizons from typical sparse data. With the wide application of 3D structural geological models in geosciences, the subsurface structural data model is being refined in terms of geometric description and expression. 3D data models of complex geological bodies can be categorized

into three types, namely, surface-based models (e.g. TIN model, GRID model, boundary representation model, line frame model, cross section model, multi-layer DEM model, etc.), body-based models (e.g. 3D grid, tetrahedron mesh, constructive solid geometry, octree model, triangular prism model, etc.) and hybrid models (Bond, 2015; Breunig, 1999; Turner, 1992). Existing data models can describe the geometric features of tectonic phenomena, but they failed to describe their geological meanings and the complex relationships among the various tectonic units. Such information is in the domain of semantics.

Semantics indicates the meaning of data as well as values (Vakarelov, 2010). Semantic description is the interpretation of an object at the semantic level, aiming to establish a connection between data and its meaning.

Semantic description has played very important roles in geographic information systems (GIS), in areas such as data sharing (Adaly et al., 2006; Zhong, 2012), integration of multi-solution model and query on heterogeneous information (Mastella et al., 2009), description of temporal succession of stratigraphy (Perrin, 2011), among other geoscience problems. On the

contrary, in the field of geological structure analysis, existing structural mode ling methods have yet to address semantics. Structural mode ling is mainly regarded as a computer graphics problem at this stage. In data analysis and processing, the implied geological semantics of data is often neglected, so semantic description is lacking in the existing description methods of tectonic phenomena. Structural mode ling often needs to overcome two obstacles:

1) The difficulty in obtaining three-dimensional spatial data, resulting in sparse and uneven distribution of data samples;

2) The complexity of spatial relationships among structural elements.

Due to these challenges, it is often too difficult for spatial geometric information alone to express geological structures accurately (Schweizer et al., 2017; Wu and Xu, 2003). Structural models established by traditional methods may not conform to geological principles and sometimes faces the problem of missing structures due to the lack of geological semantic constraints. These problems can be addressed by the semantic description of structural geological models.

In the semantic description of a structural geological model, finding out the relationships among structural elements is a very important part. Burns et al. proposed a representation method for geological topological relations by network graphs, where nodes represent spatial elements, and arcs represent topological relationships (Burns, 1981). Based on Burns' work, Samuel T. Thiele et al. put forward the concept of geological structural topology (Thiele et al., 2016). Geological structural topology is divided into three levels according to the dimensions of the spatial entities. The first order topology represents the

adjacency relationships among geological bodies; the second order topology represents the adjacency relationships among geological surfaces; and the third order topology represents the adjacency relationships among surface boundaries (see figure 1). The geological structural topology provides a preliminarily description of structural models. However, the geological structural topology is only a skeleton of structural models, with little additional structural information. Therefore it is insufficient to serve as a semantic description of structural geological models.





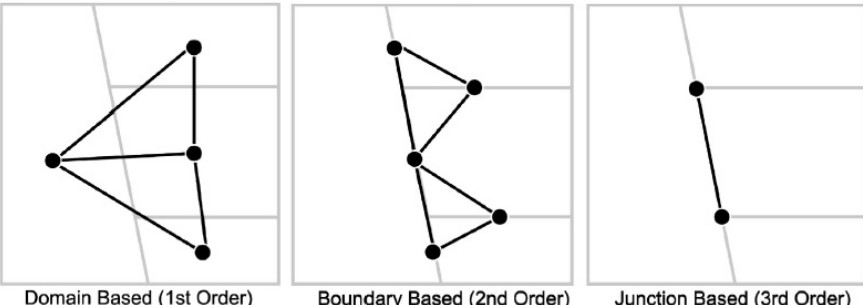

Domain Based (1st Order)    Boundary Based (2nd Order)    Junction Based (3rd Order)

**Figure 1: Examples of different orders of geological structural topology networks. The order of each topology network depends on the dimension of the geometric element represented by each node in the network. Edges in this example represent spatial adjacency (Thiele et al., 2016).**

In our study, a complete semantic description for structural geological models is proposed. Computer characterization of semantic description is carried out by using a multi-level heterogeneous network. The semantic description completely represents geometric features and structural features of structural geological models. In section 2, we put forward the concept of semantic description of structural models. In section 3, we propose the computer characterization for the semantic description. In section 4, we demonstrate the feasibility and completeness of the semantic description by proposing an

algorithm to extract the semantic description from a known structural model and an algorithm to reconstruct the model from the semantic description. In section 5, we propose a geological structural modelling method based on semantic description constraints and apply it to a survey located in China. In the last two sections, we discuss future research and draw our conclusions.

## 2 Semantic description of structural geological models

An understanding of the objective world is premised upon the ability to describe it. In recent years, the rapid development of 3D visualization technology has provided geologists with a variety of methods to analyse and process exploration data. Significant progress has been made in the direct, complete and accurate description of subsurface geological situations and resource concentration. However, there is still a lack of understanding about the meaning of such data. A geological structural model still contains mainly data instead of semantic information.

**2.1 Definition of geological structure semantics**

A description of the objective world can be characterized by entities and relationships among entities. As the basis of information, data alone does not carry sufficient meaning. Relevant data combines in certain way to form information. So we consider "data" and "relevance" to be two components of semantics (Engel and Hartmann, 1995; Xu et al., 2005). In forming semantic descriptions of the objective world, research work in image understanding has made exemplary achievements. In





the study of image understanding, it is shown that semantic descriptions of images are not only to segment and identify objects by low-order information such as texture and boundary, but also to express high-order information like how the objects in images are interrelated, and their states or the activities they involve in (Crevier, 1997; Karpathy and Li, 2015; Vinyals et al., 2015). Likewise, the semantic description of structural geological models not only needs to describe low-order

features such as geometric features, but also needs to describe high order features, such as logical relationships among structural elements. Structural elements are the basic units of geological structures and the basic components of tectonic systems. According to their origin, structural elements are divided into primary structural elements and secondary structural elements. According to geometric shape, structural elements are divided into planar structural elements (structural planes or foliations) and linear structural elements (lines or lineations).

Based on the general principles of feature analysis, a computer model of pattern recognition, namely Pandemonium Model, was proposed by Selfridge (1959). The Pandemonium Model contains 4 stages:

1) Get the figure of an object.

2) Analyse its characteristics.

3) Recognize the object at a higher level.

4) Make the right decision.

Analogous to the Pandemonium Model, for the purpose of computer information processing, our geological semantic description system is also divided into four layers: data layer, description layer, cognitive layer and application layer. The functions of each layer are as follows:

1) Data layer: The data layer stores raw data. The raw data is read from external sources and represented in the data

layer.

2) Description layer: The description layer stores entities extracted from the data layer.. The main part of this layer is to symbolize the original data (i.e. to achieve formalization). This corresponds to the procedure of extracting structural elements from geological structural model in semantic description.

3) Cognitive layer: The cognitive layer contains the logical relationship analysis among structural elements (entities

extracted in the description layer) .This layer also associates attributes with entities.

4) Application layer: The application layer consists of algorithm for operation and application of semantics according to specific tasks.

Semantic entities are the correspondence in semantic description of basic units of an object in the objective world. Multiple semantic entities and relationships among them constitute the semantic description of an object. The emphasis of this paper

on the semantic description is on the description layer and the cognitive layer: structural geological model data is abstracted into discrete semantic entities, and the relationships among these entities are analysed to realize the association between data and geological semantics.




**Definition:** the semantics of geological structures is defined as a collection of semantic entities, semantic relations (relationships among semantic entities), attributes of semantic entities and spatial geometric data. It is expressed as:

$$\textbf{\textit{GeoStruct-Semantics = \{Se, A, R, D\}}}$$

Here, the component **Se** represents semantic entities which are the basic units of semantic descriptions and the symbolic representations of objects in the objective world. Semantic entities can be divided into two categories: basic semantic entities which refer to geometric elements, and advanced semantic entities which refer to structural elements. Basic semantic entities include: points, lines, interfaces and bodies; Advanced semantic entities include: stratified structures/massive structures, planar structures and linear structures. The component **A** refers to the attributes of semantic entities. The component **R** represents semantic relations among semantic entities. Semantic relations can be divided into adjacency relations and association relations, where adjacency relations are semantic relations among the same type of semantic entities and association relations are among semantic entities of different types. The component **D** represents data, that is, the numeric information of objects in the objective world. Here, the geological structures **D** refers to the original structural data. The detailed definitions and descriptions of semantic entities, semantic relations, attributes and data are presented in the next subsection.

The semantic description of structural geological models provides additional information that is missing in traditional structural geological models. Semantic description contains the logical relationships among objects represented by data, and establishes the mapping between spatial geometric data and geological meaning. Based on this, computer algorithms can then construct structural modelling from the perspective of geoscience. In other words, existing structural modelling methods only regard the spatial data with geological meanings as ordinary spatial geometric data, so they have only solved computer graphics problems.

### 2.2 Semantic entities and semantic relations

### 2.2.1 Basic semantic entities and semantic relations

A three-dimensional structural geological model is essentially a spatial data model (Zlatanova, 2004). According to the principle of space segmentation, any complex geometric object can be represented by a finite number of simple shapes (Berlioux, 2001). Therefore, any complex geological structure can also be abstracted as a set of simple geometric shapes, which are basic semantic entities. In the last subsection, we mentioned the basic semantic entities including: bodies, interfaces, lines and points (respectively recorded as B, I, L, P). The definition symbol $A^o$ represents the interior of the basic semantic entity A, $\partial$A represents the boundary of A, and $\bar{A}$ represents the exterior of A. The interior, exterior, and boundary of the body, interface, line, and point are shown in figure 2. The interior and the boundary of the point are the point itself.





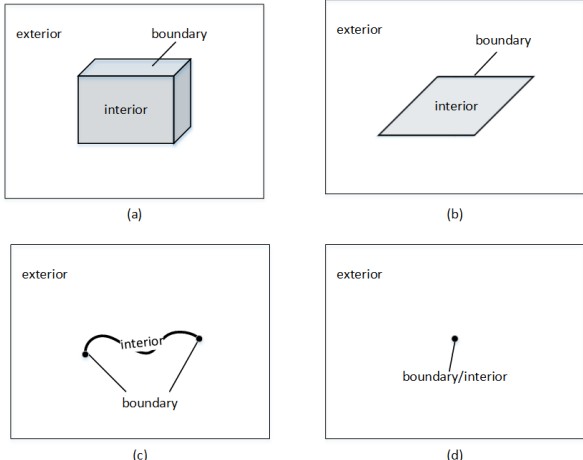

**Figure 2: The interior, exterior, and boundary of the basic semantic entities**

And there are $A^o \cup \partial A = A$, $U - A = \bar{A}$. We use the boldface **A** to represent the set. A relationship on the set **A** is defined in topology as a subset **R** of Cartesian product $\mathbf{A} \times \mathbf{A}$. $\mathbf{R} = \{(x, y) \mid x \in \mathbf{A}, y \in \mathbf{A}\}$, denoted as $xRy$. Similarly, a relation between

5    set **A** and set **B** is a subset **R'** of the Cartesian product $\mathbf{A} \times \mathbf{B}$, $\mathbf{R'} = \{(x, y) \mid x \in \mathbf{A}, y \in \mathbf{B}\}$. Therefore, we use the triple (entity$_1$, relation, entity$_2$) to represent that there is a semantic relation between semantic entity$_1$ and semantic entity$_2$, which is called a semantic unit. For the basic semantic entity, since the semantic entity is a geometric element that does not contain geological meaning, the semantic relationship between the two semantic entities A and B is actually a spatial relationship, which can be described by the 9-intersection model (9IM) proposed by Egenhofer and Herring (1990):

$$R(A, B) = \begin{Bmatrix} A^o \cap B^o & A^o \cap \partial B & A^o \cap \bar{B} \\ \partial A \cap B^o & \partial A \cap \partial B & \partial A \cap \bar{B} \\ \bar{A} \cap B^o & \bar{A} \cap \partial B & \bar{A} \cap \bar{B} \end{Bmatrix}$$

In this 3×3 matrix, the items with empty intersections are set to 0, and the items with non-empty intersections are set to 1. This can distinguish $2^9 = 512$ kinds of spatial relationships. Most of them have no practical meaning. And 31 of these are the fundamental relationships we have studied and used in this paper, including the association relations among basic semantic entities:

15    1)    Points to lines: ((boundary) point, *composition*, line), ((internal) point, *composition*, line).

       2)    Lines to interfaces: ((external) line, *composition*, interface), ((internal) line, *composition*, interface).

       3)    Interfaces to bodies: ((external) interface, *composition*, body), ((internal) interface, *composition*, body).

       The adjacency relations among basic semantic entities including:

       1)    Points to points: (point, *disjoint*, point), (point, *equal*, point).

20    2)    Lines to lines: (line, *disjoint*, line), (line, *intersect*, line), (line, *overlap*, line), (line, *equal*, line).



3) Interfaces to interfaces: (interface, *disjoint*, interface), (interface, *overlap*, interface), (interface, *equal*, interface), (interface, *intersect*, interface), (interface, *cover*, interface), (interface, *coveredby*, interface).

4) Bodies to bodies: (body, *disjoint*, body), (body, *meet*, body), (body, *equal*, body), (body, *overlap*, body), (body, *cover*, body), (body, *coveredby*, body), (body, *contain*, body), (body, *inside*, body).

5   The graphic descriptions of these semantic relationships and the 9-intersection models are shown in figures 3 and 4. Basic semantic entities divide structural geological models into a variety of units from geometric point of view, and describe spatial topological relations among these parts of geological models by relations among the geometric elements. The geometric elements of interface, line and point have attributes to distinguish whether they are internal or external to the upper level entity. The element body has no attributive information. Here data information refers to spatial geometric data.

10  The complete basic semantic description system is shown in Table 1.

**Figure 3: Semantic association relations of basic semantic entities.**





**Figure 4: Graphic descriptions of semantic adjacency relations of basic semantic entities.**

| Semantic entity | | Semantic relation | | Attribute | Data |
|---|---|---|---|---|---|
| **Name** | **Explanation** | **Adjacency relation** | **Association relation** | | |
| Body | An object with a certain volume in 3D space. | Disjoint Equal Meet Overlap Contain Cover Coveredby Inside | | The type of entities: exterior or interior | Spatial geometry data |
| | | | Interfaces compose bodies. | | |
| Interface | An object with a certain area but without volume in 3D space. | | | | |
| | | | Lines compose interfaces. | | |
| Line | An object with a certain length but without volume and area in 3D space. | | | | |
| | | | Points compose lines. | | |
| Point | An object without volume, area and length in 3D space. | Disjoint Equal | | | |

Table 1: Basic semantic entities and their corresponding semantic relations and data.

#### 2.2.2 Advanced semantic entities and semantic relations

The basic semantic description describe structural geological model based on geometric shapes. , The advanced semantic description takes structural elements as semantic entities and describes structural geological models based on geological concepts. The former is close to the way of computers cognitive structural models while the latter is close to the way of human cognitive structural models.

A 3D geological model consists of a set of geological structural elements, include stratified structures (sedimentary rocks) or massive structures (igneous rocks and other geological blocks without obvious occurrence), planar structures (foliations, fault planes and joint planes) and linear structures (considering the scale of the structural model, most of lineations will not be described in the model, so here linear structures only refer to intersection lineations of two structural planes and large lineations like boudinages and mullions). As we know, the assemblage of structural elements is the result of their geological history. Each definite structural element of the model is the result of a significant tectonic process. Here tectonic processes refer to geological processes that cause the formation or deformation of structural elements. According to geological events and the changes in the rock mass they caused, tectonic processes can be divided into three categories: rock mass generation, rock mass destruction and rock mass deformation (see figure 5 and figure 6). Rock mass generation represents geological processes in which other non-rock mass materials are converted into rock masses. This include magma condensation (such as the formation of the crust), sedimentation, crystallization, magma squirting, magna intrusion, cementation and extraterrestrial





material. Rock mass destruction refers to geological processes of transforming rock mass into non-rock mass material (erosion, weathering and melting). Rock mass deformation corresponds to geological processes with only the shape or volume of the rock mass changes (faulting, folding, bioturbation and compaction). However, some tectonic events have very similar results (like erosion and weathering) or they are two directions of the same process (like compaction and extension),

5  we merge these tectonic processes. The classification of structures and the tectonic processes that form them is shown in figure 7. Moreover, as shown in figure 8, tectonic events can correspond to structural elements.

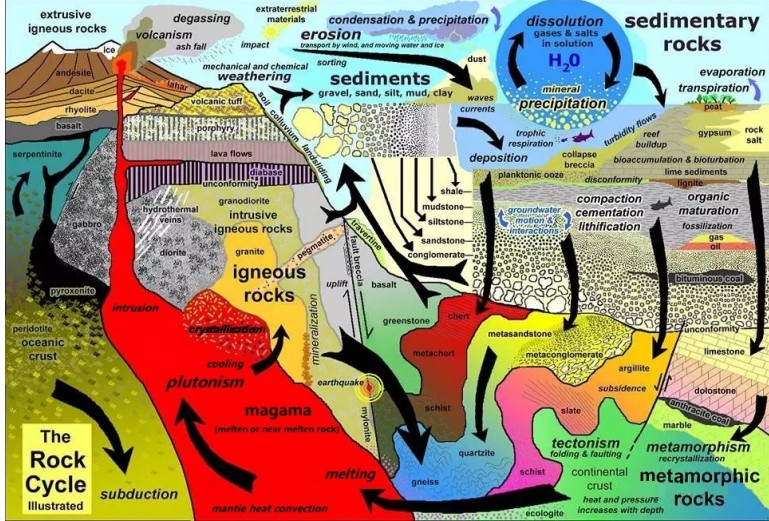

**Figure 5. Tectonic processes and the rock cycle illustrated**



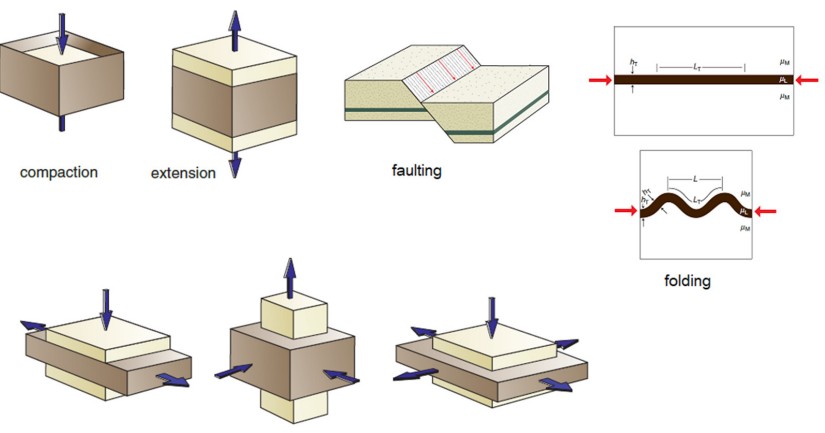

**Figure 6. Some structural deformation (Fossen, 2016).**

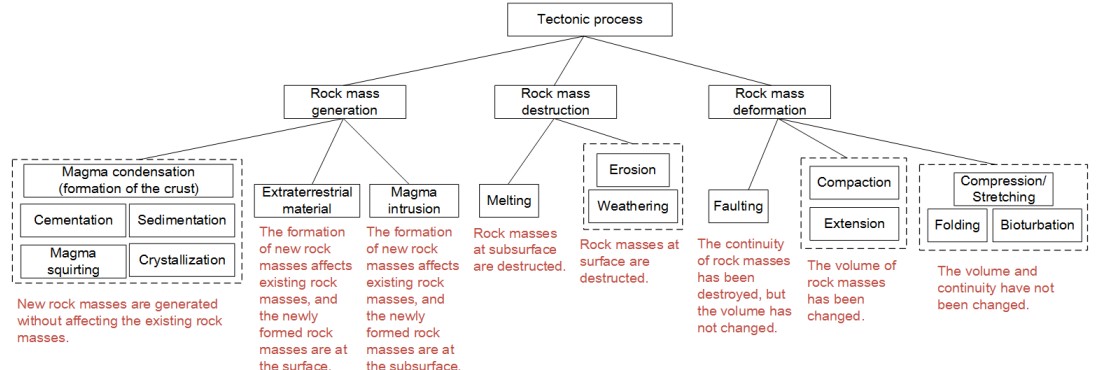

**Figure 7. Tectonic process classification and classification basis (red words in the figure).**



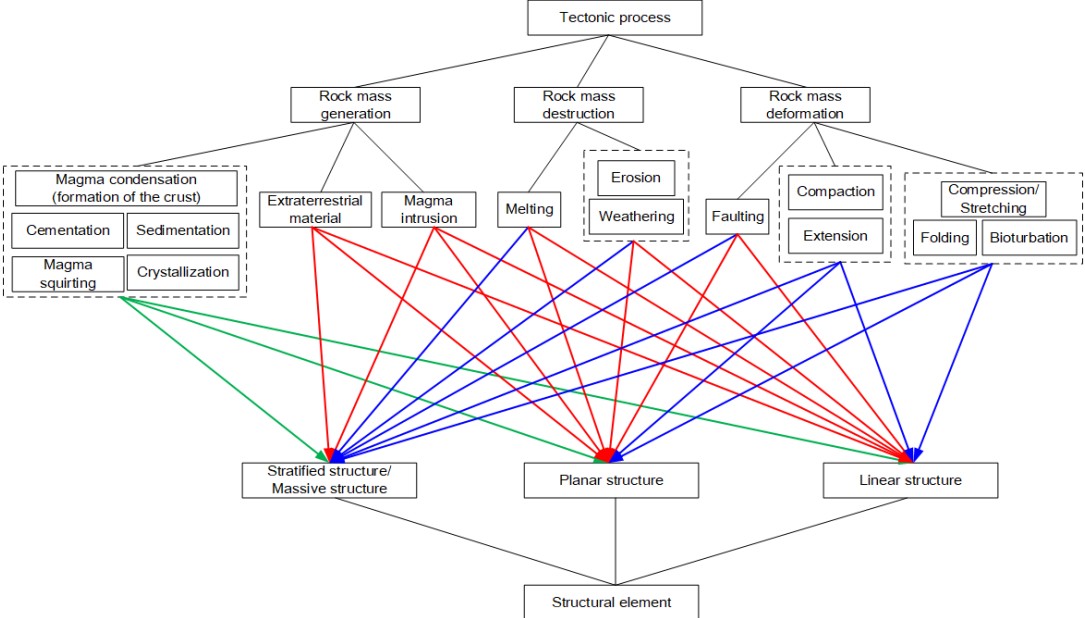

**Figure 8. Relations between tectonic processes and structural elements. The green links indicate that the tectonic process forms corresponding structural elements. For example, the process sedimentation creates a new stratum (stratified structure) and also produces a stratigraphic plane (planar structure). When a stratum thins out, a thin out line (linear structure) will be created. The blue links mean the geologic event deforms existing structural elements. The red link is a mixture of the first two: a new structural element is created while changing an old structural elements.**

Advanced semantic entities in the semantic description are actual structural elements (stratified structures/massive structures, planar structures and linear structures). While basic semantic entities only have geometric meaning, advanced semantic entities emphasize the geological meaning, so the semantic relations between the advanced semantic entities not only need to describe the spatial topological relations between two structural elements, but also need to describe the geological meaning of the adjacency relationship. The relationship between structural elements is determined by tectonic events. Because of the occurrence of a geologic event, some structural elements were formed, and the newly created element is adjacent to existing structural elements. It should be noted that in structural geology, there is no corresponding association relation concepts between two disjoint structural elements. Therefore, we only need to discuss adjacency relations among advanced semantic entities.

Semantic adjacency relations are defined by the tectonic event creating the structural elements itself and the nature of tectonic events determines the characteristic of the interface between the adjacent structural elements. For stratified structures/massive structures, there are three types of geologic events that have red or green links to stratified structure/massive structures. They form stratified structure/massive structure elements. Other six types of events that have



red or green links to planar structure element deform and create boundary of stratified structures/massive structures—planar structures. So according to the semantic unit (entity$_1$, relation, entity$_2$) we mentioned before, there are total $3\times6\times3=54$ possible adjacency relationships between structures/massive structures. For planar structures, there are five events that form the element (sedimentation, magmatic intrusion, erosion, faulting and compression/stretching) and seven events that deform

or create linear structures (boundaries of planar structures). Therefore there are $6\times6\times6=216$ possible adjacency relations among planar structures. For linear structures, in theory, boundaries of them are the end points of lines. However, in the terminology of structural geology, there is no corresponding concept as point structures. So "points" here only exists in the raw data represented by discrete points in 3D space. So when we discuss about adjacency relationships among linear structures, the relations are determined by geologic events that affect linear structures themselves. The number of all possible

relationships among linear structures is also $6\times6\times6=216$. Some relationships may not have geological meanings or corresponding geological concepts because we have yet to find any real instances of them. Some do not need to be distinguished in details or they express the same geological concept, this part can be merged into one semantic relation. This semantic relationship definition method can cover structural geological concepts that have not yet been defined.

 According to the concepts of structural geology, we use ten common relationships to describe semantics. They are divided

according to the types of semantic entities:

  1) Stratified structures/massive structures to stratified structures/massive structures: (stratified structure, {*conformable, unconformable, intrusive, sedimentary, fault*｝ *contact*, stratified structure).

  2) Planar structures to planar structures: (planar structure, {*stagger, limit, cut, mutually stagger, trace*}, planar structure).

3) Linear structures to linear structures: (linear structure, *reform*, linear structure).

 A *conformable contact* is one in which the strata are in unbroken sequence and in which the layers are formed one above the other. A *unconformable contact* is a surface of erosion or nondeposition that separates younger strata from older strata. An *intrusive contact* is a rock, magma, or sediment mass that has been emplaced into another distinct unit. A *sedimentary contact* is weathering and erosion of igneous intrusions and then covered by new sedimentary rocks. *Fault contact* is a

structural contact, that is, the interface between the intrusive rock mass and the surrounding rock is the fault plane. A *stagger* relation means a planar structure formed later cutting off a surface structure formed earlier. A *limit* relation means when a planar structure grows to another planar structure, the younger surface is terminated by the older surface and the younger one does not pass through the older one. A *cut* relation actually has the same spatial relationship with the *limit*, but it means the younger surface cut out the older surface. A *mutually stagger* relation refers to two planar structures intersect and mutually

cut each other, and the two form a conjugated relation. A *trace* relation means the planar structure grows along a formed planar structure. For example, a tension joint may be formed along a group of shear joints. A *reform* relation means the younger linear structure changes the shape of the older linear structure and the transformation. Generally, the younger linear structure will destroy the continuity of the older linear structure.





While emphasizing the geological meaning of structural elements in advanced semantic description, we must not ignore the geometric relations between structural elements. The same semantic relationship may have completely different topologies in the actual model, spatial geometry will help us to clarify semantic relationships. The 9-intersection model is a good mathematical model for describing the spatial topology. However, different geological relationships may also have the same

spatial topology. For example, the geometric spatial topological expressions of conformable contact and parallel unconformable contact of two adjacent stratum are the same. So the basic 9-intersection model is insufficient for our requirements.

We know that geological structures can be divided into primary structures and secondary structures according to the chronological order of formation. The primary structure is a structure formed during the formation of rock mass, and the

secondary structure is a geologic deformation caused by geologic events after rock formation. Geometrically, a geological structure is composed of its interiors and its boundaries. Due to some tectonic processes, part of the original boundary and original interior of structural elements may be destructed or deformed. Based on the concept of primary/secondary structure, we divide both boundaries and interiors into primary ones and secondary ones. We mentioned that advanced semantic relations are defined by the nature of entities and their interface. We define the primary boundary and interior as: the

boundary and interior generated by the tectonic events that generated the structural element they belong to and not changed by subsequent tectonic events. The "change" here includes both destruction and deformation. In contrast, secondary boundaries and interiors are the boundaries and interiors that have been changed by tectonic events after the formation of the structural elements to which they belong or that are not generated by the tectonic event that formed their structural element. We give the graphic explanations of semantic relations among stratified/massive structures in figure 9 to illustrate how we

decide whether boundaries and interiors are primary or secondary. The corresponding tectonic events to structural elements and interfaces, and the attributes of boundaries and interiors are as follows: (a) Conformable contact: A--sedimentation, B--sedimentation, interface--sedimentation. The nature of interface is decided by the tectonic event that generates B(A), and according to figure 8 sedimentation does not change any existing structural elements, so the boundaries and interiors of both A and B are primary. (b) Unconformable contact: A--sedimentation, B--sedimentation, interface--erosion. According to

figure 8 the event erosion creates a new planar structure and deforms the existing structural element A, so the boundary is secondary to both A and B, and the interior of A is also secondary. (c) Intrusive contact: A--sedimentation, B--intrusion, interface--intrusion. Similarly, the interface is a primary boundary to B and a secondary boundary to A. And the interior of A is secondary. (d) Sedimentary contact: A--intrusion, B--sedimentation, interface--erosion. The interface is a secondary boundary to A and B. The interior of A is secondary. (e) Fault contact: A--sedimentation, B--intrusion, interface--faulting.

The boundaries and interiors are all secondary.  In the same way we can decide whether the interiors and boundaries of planar structures and linear structures are primary or secondary.





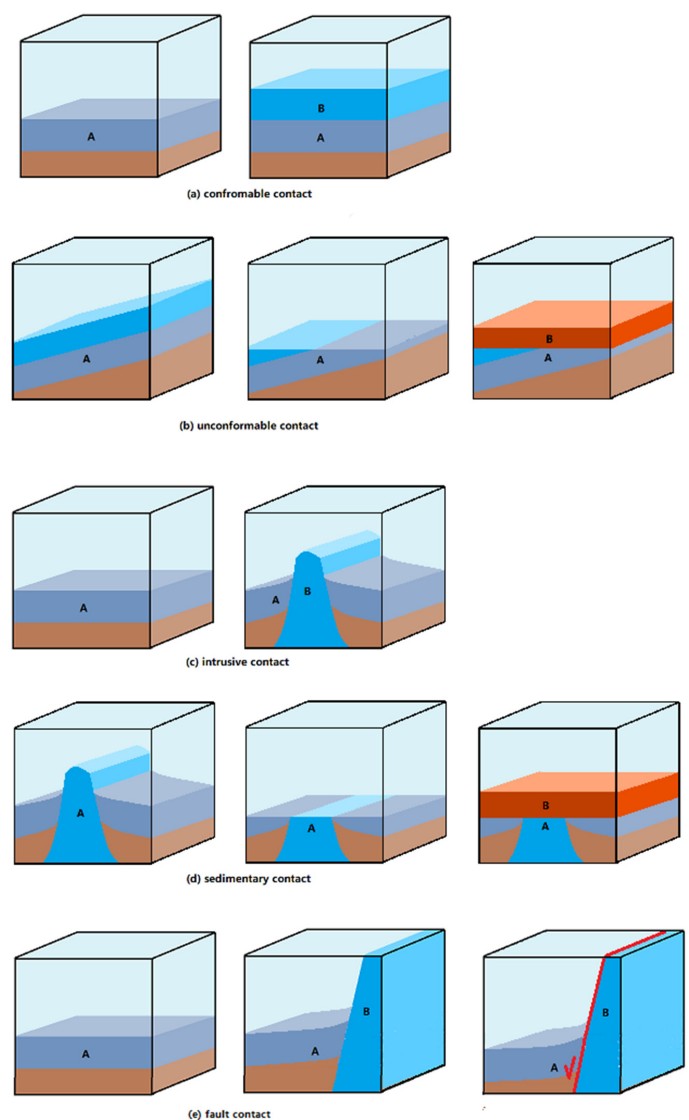

**Figure 9. The graphic descriptions of semantic relations among stratified/massive structures.**



As we mentioned before, in the 9-intersection model, 0 is for empty sets and 1 is for nonempty sets. The symbol $\partial A$ represents the boundary and $A^o$ represents the interior of the element A. In order to distinguish more geological relations, we define an extended 9-intersection model: the terms whose intersections are empty in the matrix are set to 0, and for the terms with nonempty intersections, if the intersection contains secondary boundaries or interiors of elements then the term is set -1,

5   the other nonempty term is still set 1.

The extended 9-intersection model and graphic descriptions of the 11 semantic relations are shown in figure 10. We can see in figure 10 that one geological relation can have multiple geometric assemble way, like stagger, limit and reform. There is no association relation among these three structural elements in structural geology. The attribute of stratified structures/massive structures is the geological time of their formation. The attributes of planar structures and linear structures

10  are their specific structural types. The data here means the occurrence information of structures, such as strike, dip, thickness, etc. The complete advanced semantic description system is shown in Table 2.

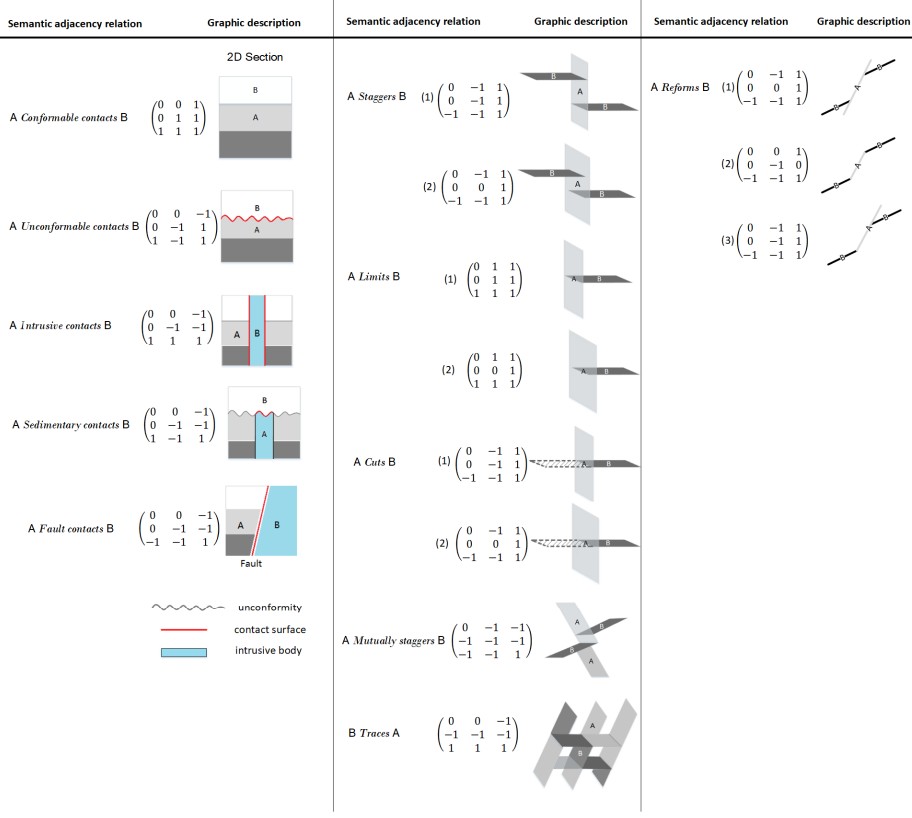

**Figure 10. Graphic descriptions of semantic adjacency relations of advanced semantic entities.**





| Semantic entity | | Semantic relation | | Attribute | Data |
|---|---|---|---|---|---|
| Name | Explanation | Adjacency relation | Association relation | | |
| Stratified structure / Massive structure | A layer or group of rock masses formed during a certain geological time. | Conformable contact Unconformable contact Intrusive contact Sedimentary contact Fault contact | | The geological time of formations. | The occurrence of structures. |
| Planar structure | Horizons, fault planes, intrusion contact planes, joint planes are included. | Mutually satgger Stagger Trace Limit Cut | | The specific type of structures. | |
| Linear structure | Intersection lineations of two structural planes and large lineations. | Reform | | | |

Table 2. Advanced semantic entities and their corresponding semantic relations, attributes and data.

The basic semantic description and the advanced semantic description describes structural geological models based on the geometric meaning and the geological meaning of data, because the two describe the same object, therefore, there must be correlations between the two semantic descriptions. We found that bodies, interfaces, lines of basic semantic entities can be

5    several-to-one mapped to stratified structures/massive structures, planar structures, linear structures of advanced semantic entities. For example, a stratum that is cut through by a fault may still be a continuous whole in space, but it can be logically regarded as the fault cutting the stratum into two bodies. Similarly, a fault plane that cuts through strata and intersects with horizons is still a continuous geological surface, but can be logically seen as being cut into multiple interfaces by the intersection lines. A boudinage is originally composed of a few separated linear geological bodies, so it can be regarded as

10   multiple lines (see figure 11). This kind of mapping is between different types of semantic entities, so it is also an association semantic relation. This kind of relationship is the key to assign geological meaning to geometric data. Together with the association relationships in the basic semantic description, we can query the structural elements and then get geological meanings of each spatial discrete point. Therefore, the combination of basic semantic description and advanced semantic description enables the expression of geological semantics with computers.





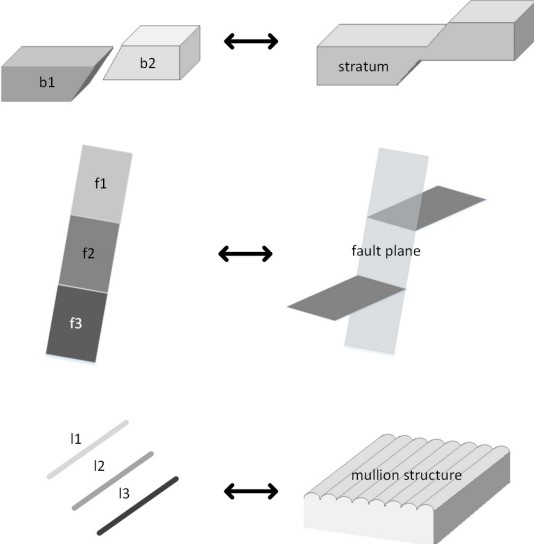

**Figure 11. Mapping between basic semantic entities and advanced semantic entities. For example, a stratum can correspond to multiple bodies because of the cutting of faults, a fault plane can correspond to multiple interfaces because of the cutting of the intersections with horizons, and a mullion structure can correspond to multiple lines.**

## 3 Computer Characterization of Structural Model Semantics

Our semantic description of a structural model aims to bridge between geoscience and information science. Therefore, it needs a suitable computer representation. This would also help the application of artificial intelligence to the field of geoscience.

From the semantic description system in the previous section, we can see that semantic entities are divided into two types: basic semantic entities and advanced semantic entities. They are essentially geometric elements and structural elements. Basic semantic entities are divided into 4 layers and high-level entities can be composed of low-level entities. A semantic relationship between the same kind of entities is an adjacency relationship, and a semantic relationship between different kinds of entities is an association relationship. The mapping between stratified structures/massive structures, planar structures and bodies, interfaces, lines links the two semantic descriptions, that is, to link geological meanings with spatial data.

Based on the characteristics of the semantic description mentioned above, we proposed a multi-level heterogeneous network as a computer characterization of the semantic description of structural geological models. We express semantic descriptions by semantic entities and semantic relations among entities. As a network graphs happens to have the two elements "entities" and "relationships", and the formal expression of network graphs is simple, we therefore use network diagrams as computer



characterizations of the semantic description. Network nodes represent the abstractions of semantic entities, and the direction of the arc is used to reflect primary and secondary relations in the semantic relation. Annotations on an arc represent specific types of semantic relationships between the two connected nodes. The triple **(node1, arc, node2)** forms a basic network element to indicate that there is a semantic relationship between node1 and node2. Nodes in the same layer of network

5    represent the same kind of entities. The structure of a network is shown in figure 12. Both basic semantic description (points, lines, interfaces, bodies) and advanced semantic description (stratified structures/massive structures, planar structures, linear structures) can be used to construct a structural model.

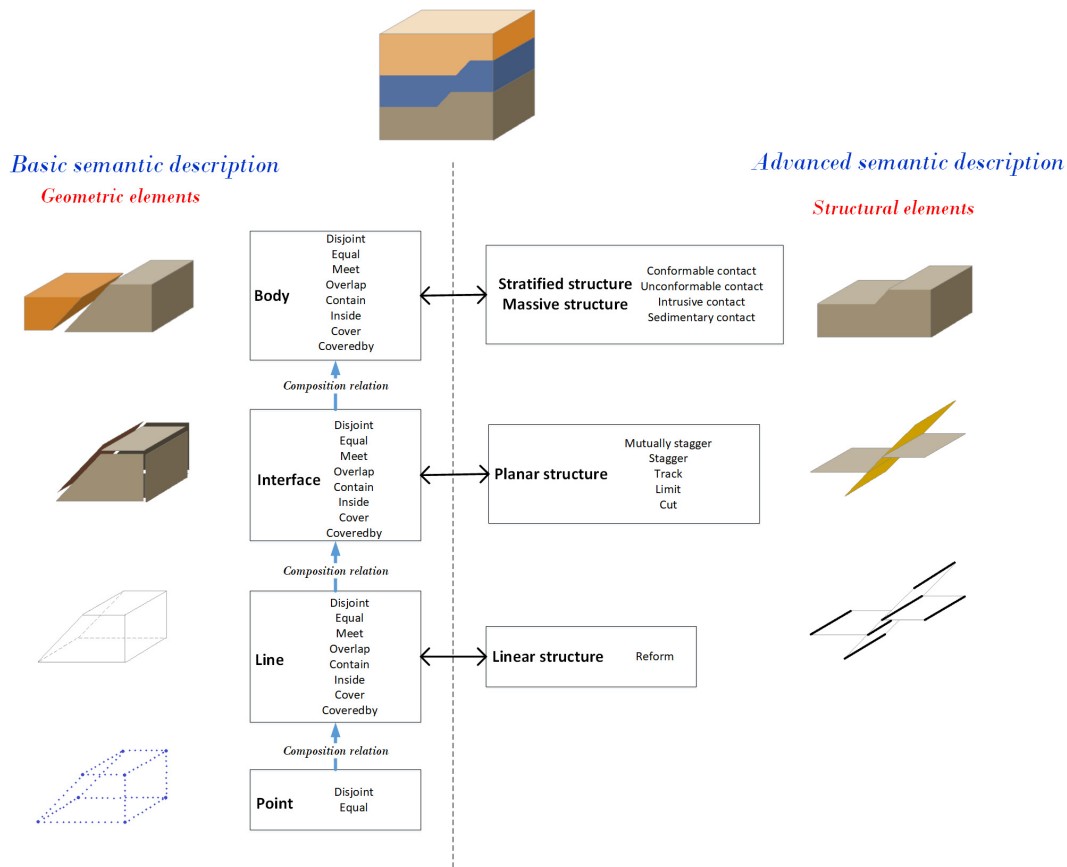

**Figure 12. The schematic diagram of a multi-level complex heterogeneous network structure where the network is a computer**
10   **characterization of semantic descriptions of structural geological models. The solid double arrows indicate the mapping between stratified /massive structures and bodies, planar structures and interfaces, linear structures and lines.**





The semantic description of structural geological models encompass both geometric shape information and structural elements information of the models. With existing modelling methods, computer algorithms can only display the structural model. The semantic description enables computer algorithms to recognize structural geological models similar to human cognition.

## 4 Methods

In this section, we propose two methods. The first method automatically extracts semantic descriptions from known 3D structural geological model data. The second method reconstruct the structural geological model based on semantic description. The extraction method proves the feasibility of our proposed semantic description system, and the reconstruction method proves the completeness of semantic description in information representation. We use italics when we express semantic relations in this chapter.

### 4. 1 Extraction of semantic descriptions from structural geological models

A process for automatic extraction of semantic description is proposed in the case of structural model data, which illustrates the feasibility of semantic description. The method is divided into three parts, which extract the basic semantic description, the association between two kind of semantic description and the advanced semantic description, respectively.

### 4. 1. 1 The input and output

The known information comes from seismic interpretation results, including structural interpretation, stratigraphic geochronology and spatial point coordinates. The input data is a structural geological model that has been established. The geological bodies and sub-surfaces have been identified in the known structural model. Our structural model data includes three parts, the first part records the sub-surface IDs that make up the geological body. Then the second part records the micro-topology of points that form the triangulated mesh of each sub-surface. And the third part records the coordinates for each point.

The output is semantic relations represented by relational tables. The attributes in the advanced semantic description are attached to the semantic entities according to the geologist's interpretations of the data. The data part we defined in the semantic description forms a data set separately and is not directly represented in the output for visualization reason.

### 4.1.2 Steps of extracting semantic descriptions

The first step of the extraction is to obtain the basic semantic description which is based on basic semantic entities and their semantic relations.

**Part 1.Basic semantic descriptions**:

***Input***:***Three-dimensional structural geological model data***



*Output:Semantic relations and attributes of basic semantic entities*

1. Traverse geological bodies. Geological bodies correspond to body semantic entities. Both interior and exterior sub-surface of geological bodies correspond to interface semantic entities and these interfaces have the association relation *compose* with the body. So association relations among interfaces and bodies can be obtained. Go to step 2.

2. If there is one interface associated with two bodies at the same time, then there is an adjacency relation between the bodies. So adjacency relations of bodies can be obtained by comparing the interfaces associated with the bodies. The interface entity that is only associated with one body entity and not on the boundary surfaces of the structural model is an interior interface. The rest are exterior interfaces. Go to step 3.

3. Traverse the interface entities and compare the edges of any two interfaces. If two interfaces have the same edge, then
there is an adjacency relation between the two interfaces. The same edges that can be connected end to end is a line entity. Edges at the boundary surface of the structural model that are not shared with other interfaces can also be connected end to end to form a line entity. Therefore adjacency relations among interfaces and association relations from lines to interfaces can be obtained. The line entity that is not on the boundary of the interface is an interior line and others are exterior lines. Go to step 4.

4. According to the edges that constitute the line entity, the points on the line entity, that is, the point-to-line association relations, can be found in the input data. The point that only belongs to one edge of a line entity is an exterior point otherwise is an interior point.  Go to step 5.

5. Traverse the line entities, and compare the points associated with any two line entities. There is an adjacency relation between the line entities that have a same point. Adjacency relations of lines are obtained. Go to step 6.

6. All bodies compose the structural model together. Therefore there are association relations from every bodies to the model.

The advanced semantic entities have been reflected in the interpretation results, so we are going to find the mapping between the basic and advanced semantic entities in the second step, that is, the relationships between the two kinds of semantic descriptions.

**Part 2. Associations between the basic and advanced semantic description:**

*Input:Structural interpretation and stratigraphic geochronology from the seismic data, basic semantic descriptions.*

*Output:Associations between basic semantic entities advanced semantic entities*

1. Extract association relations among geological bodies and stratified structures/massive structures based on the geological time information. We believe that geological bodies formed in the same geological period belong to the same
strata. Move to step 2.

2. An interface entity is essentially a part of a structural plane (planar structure). We can find the structural plane of an interface by comparing coordinates of points and then get association relations among interfaces and planar structures. Move to step 3.





3. A line entity is also a part of a linear structure. Also by comparing coordinates we get association relations among lines and linear structures.

In the third step, we automatically judge the semantic relations between the advanced semantic entities according to the spatial topological relationships between the basic semantic entities and the mapping relationship between the two types of semantic entities in the second step.

**Part 3.Advanced semantic description:**

*Input:Structural interpretation from the seismic data, advanced semantic entities.*

*Output:Semantic relations and attributes of advanced semantic entities.*

1. According to the structural planes where stratified structures/massive structures contact, we can find adjacent stratified structures/massive structures. The specific structure type of the contact plane determines the type of the adjacency relation. Move to step 2.

2. Determine adjacency relations among planar structures. Move to step 3. The determinants of adjacency relations are as following (see figure 13.):

    (a)*Stagger*: the structural plane $S_1$ intersects with the structural plane $S_2$ (interface $f_1$ from $S_1$ *meets* with interface $f_5$ from $S_2$) and $S_1$ is continuous ($f_1$ *meets* with $f_2$ and $f_2$ *meets* with $f_3$) while $S_2$ is discontinuous (because $f_4$ is *disjoint* with $f_5$). Then $S_2$ *staggers* $S_1$.

    (b)*Limit* and *Cut*: $S_1$ intersects with $S_2$ and only one interface f1 from $S_2$ *meets* with $S_1$. Then $S_1$ *limits* $S_2$.

    (c)*Mutually stagger*: $S_1$ intersects with $S_2$; $S_1$ is discontinuous while $S_2$ is discontinuous. Then $S_2$ *mutual cuts* $S_1$.

    (d)*Trace*: all interfaces from $S_2$ ($f_1$, $f_2$ and $f_3$) *equals* or *coveredby* or *inside* some interfaces from $S_1$ ($f_4$, $f_5$ and $f_6$). Then $S_2$ *tracks* $S_1$.

3. Determine adjacency relations among linear structures. If lines from different linear structures have adjacency relations *meets*, which means that the continuity of one linear structure is destroyed by the other linear structure, then among these linear structures have the adjacency relation *reform*.





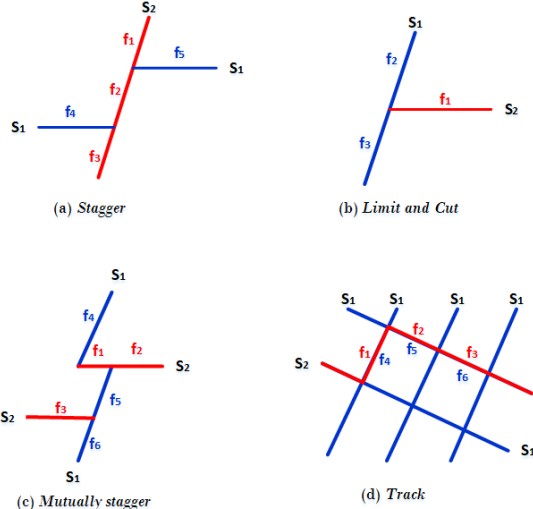

**Figure 13. (a)(b)(c)(d)The adjacency relation stagger, limit, mutual cut and track between planar structures and relationships among interfaces under these adjacency relationships of planar structures.**

We obtained the structural geological model data of a region of Xinjiang, China (see figure 14). We extracted the semantic

5    description of the structural model of this region according to the above method. There are 16 geological bodies (namely b0, b2,…, b15) from 9 strata or rock masses (namely S1, S2,…,S9), 43 interfaces (namely s0, s1,…, s42) from 11 planar structures (namely H1,…,H6, U, F2, F2-1, F3, F4, respectively represent the horizons, the unconformity plane and the fault planes) together with the top surface and the bottom surface of the structural model, 52 lines (namely l0, l1,…,l51) from 31 linear structures (namely L1, L2,…,L31), and 984 points on the lines.





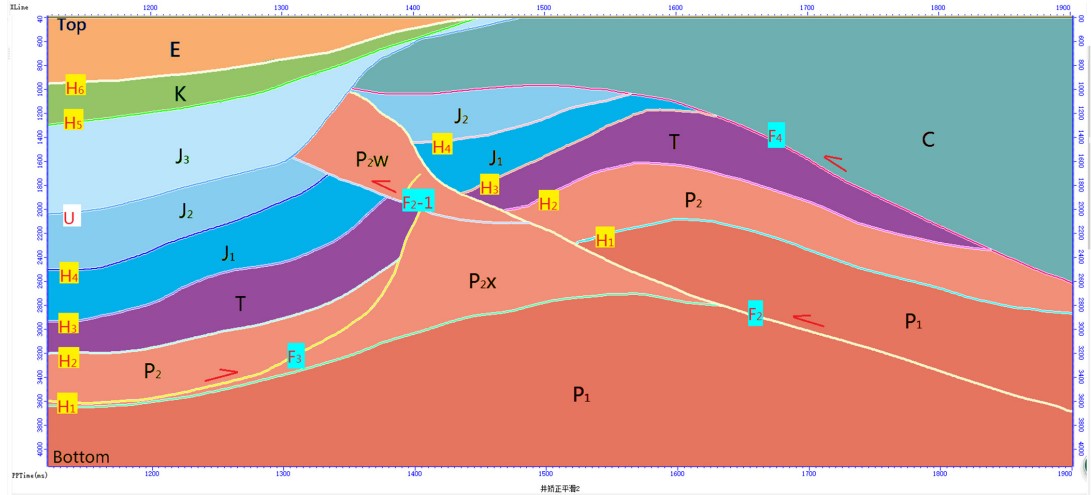

**Figure 14. A cross section of 3D structural geological model in a certain area of Xinjiang, China. Horizons, the unconformity plane and fault planes are respectively labelled with H, U, and F. Each geological body unit is tagged with the geological time of its formation. Colouring follows the Commission for the Geological Map of the World (Cohen et al., 2013).**

5   The semantic description is represented as a complex network laid out in three-dimensional space according to the computer characterization method proposed in section 3, and is divided into multiple levels according to the class of semantic entities (see figure 15). The nodes in one layer represent the same semantic entity. It should be noted that we have not shown all *disjoint* relations in the network because entities without other semantic relations naturally have a *disjoint* relation. Similarly, in order to make the visualization of the network clearer, we also merged semantic entities with *equal* relationships. The

10   main ones have equal relations are lines and point entities. Lines are duplicated because when multiple interfaces intersect at one location, a two-to-two intersection creates a line entity. Therefore, the points are repeatedly recorded because the lines have duplicates, but in fact, the parts of two lines that overlap or intersect are the same points. So it is reasonable to merge entities with *equal* relations into one entity. Of course we also merge actually the same semantic relations and keep different ones.




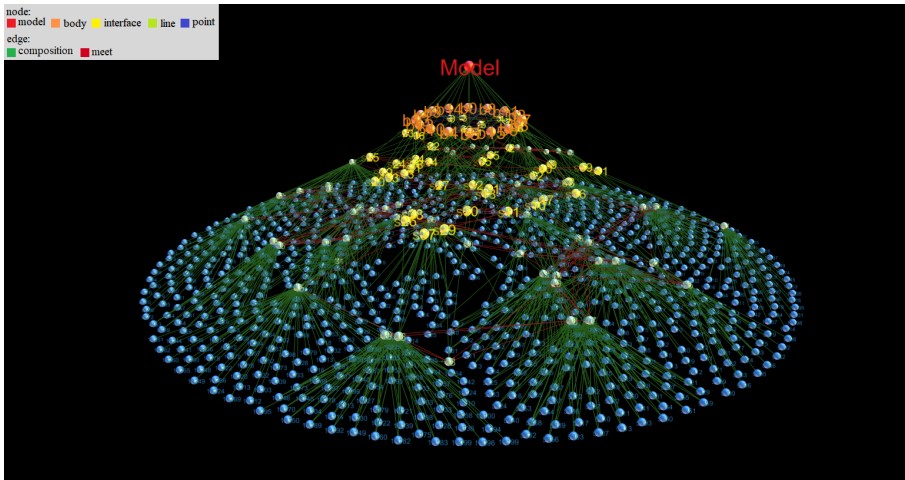

**(a)** Basic semantic description network.

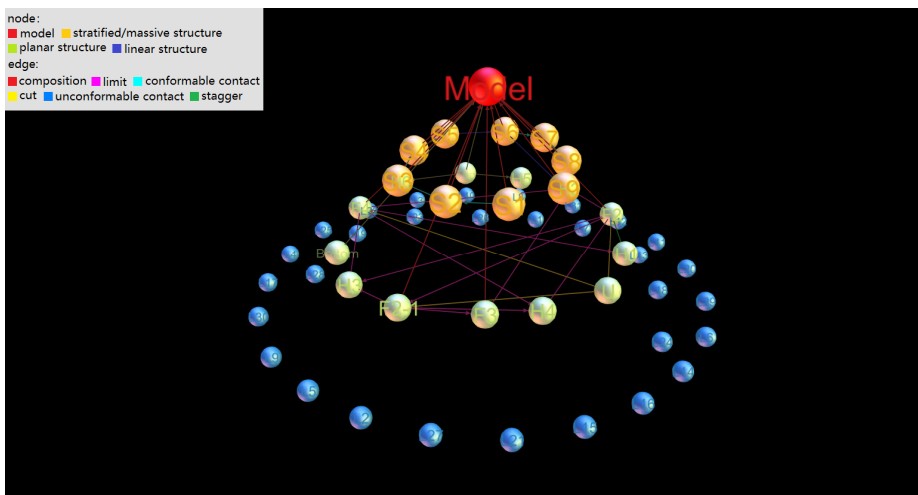

**(b)** Advanced semantic description network.



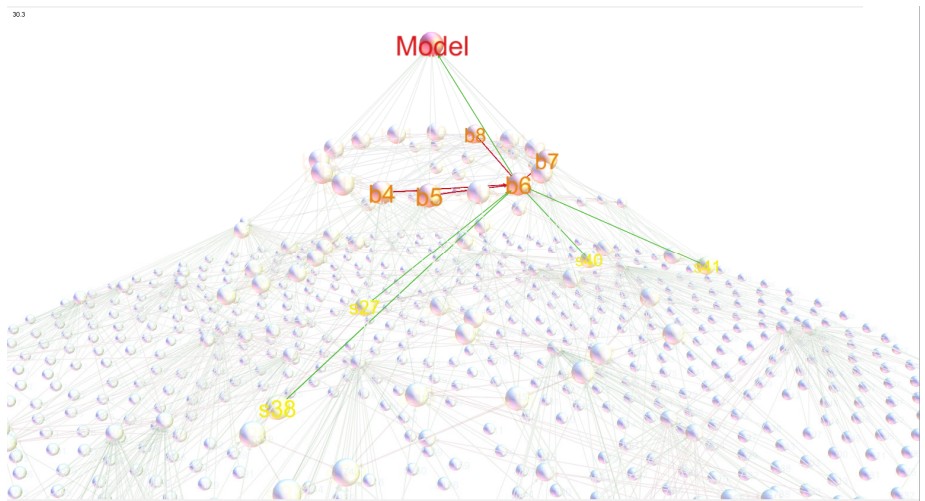

**(c)**  **A part of the basic semantic description network.**

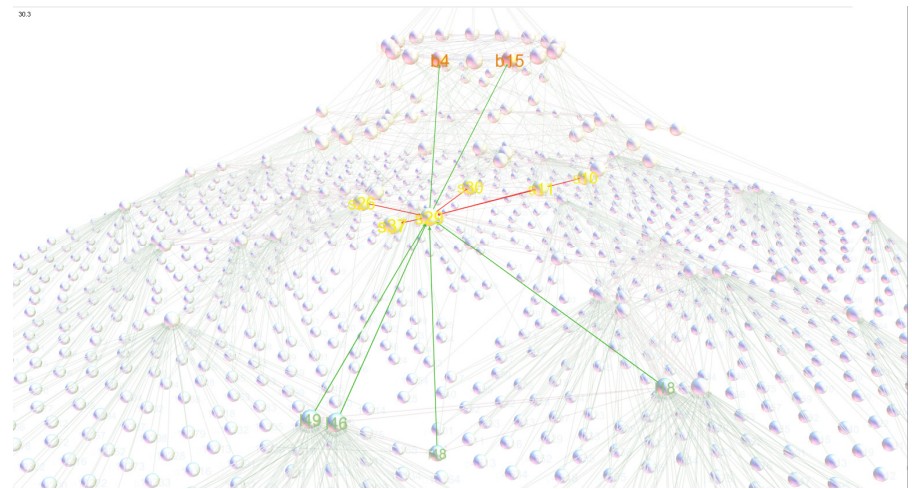

**(d)**  **Another part of the basic semantic description network.**

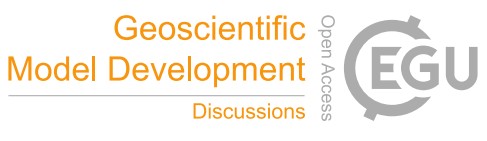

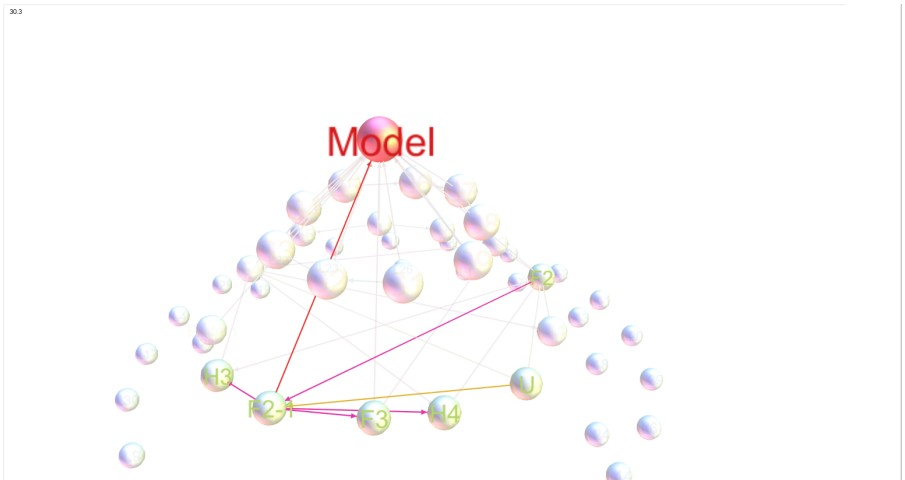

**(e)  A part of the advanced semantic description network.**

**Figure 15. (a) Complex network representation of the basic semantic description. (b) Complex network representation of the advanced semantic description. The colour of nodes represents the kinds of semantic entities represented by nodes, and the colour**
5 **of edges represents the kinds of semantic relations. (c)(d)(e)Three parts of the semantic description network. Through software interaction, it is possible to clearly observe the connection of each node in the network with other nodes. For example, from (c), it can be seen that there are 25 points on the line l40, and l40 meets with the other 7 lines l32, l35, l38, l40, l41, l43, l44, and belongs to the two interfaces s25 and s21. (d) The body b13 is composed of interfaces s15, s20, s21, s25, s34, s35, and meets bodies b14, b12, b11, b4 and b8. (e)The fault plane F2 limits horizons H2, H3, H4, the unconformity plane U and another fault plane F2-1, and**
10 **staggers the horizon H1. This figure was generated by Gephi v0.8.1 beta.**

The association relations between the basic semantic description and the advanced semantic description is expressed in the

form of relational tables (see table 3, 4 and 5.).

| Body | Stratified structure/ Massive structure | Attribute: geological time |
|------|------|------|
| b0 | S8 | E |
| b1 | S7 | K |
| b2 | S6 | $J_3$ |
| b3 | S5 | $J_2$ |
| b4 | S2 | $P_2$ |
| b5 | S4 | $J_1$ |
| b6 | S3 | T |
| b7 | S2 | $P_2$ |
| b8 | S2 | $P_2$ |
| b9 | S1 | $P_1$ |
| b10 | S4 | $J_1$ |
| b11 | S9 | C |
| b12 | S3 | T |



| | | |
|---|---|---|
| b13 | S2 | P2 |
| b14 | S1 | P1 |
| b15 | S5 | J2 |

**Table 3. Mapping between bodies and stratified/ massive structures.**

| Interface | Planar structure | Attribute: type | Interface | Planar structure | Attribute: type |
|---|---|---|---|---|---|
| s0 | top | Boundary | s22 | F2 | Fault |
| s1 | top | Boundary | s23 | F2 | Fault |
| s2 | top | Boundary | s24 | F2 | Fault |
| s3 | top | Boundary | s25 | F2 | Fault |
| s4 | H6 | Horizon | s26 | F2-1 | Fault |
| s5 | H6 | Horizon | s27 | F2-1 | Fault |
| s6 | H6 | Horizon | s28 | F2-1 | Fault |
| s7 | H5 | Horizon | s29 | F2-1 | Fault |
| s8 | U | Unconformity | s30 | F2-1 | Fault |
| s9 | U | Unconformity | s31 | H4 | Horizon |
| s10 | U | Unconformity | s32 | H4 | Horizon |
| s11 | F4 | Fault | s33 | H3 | Horizon |
| s12 | F4 | Fault | s34 | H2 | Horizon |
| s13 | F4 | Fault | s35 | H1 | Horizon |
| s14 | F4 | Fault | s36 | H1 | Horizon |
| s15 | F4 | Fault | s37 | H4 | Horizon |
| s16 | F4 | Fault | s38 | H3 | Horizon |
| s17 | F2 | Fault | s39 | F3 | Fault |
| s18 | F2 | Fault | s40 | F3 | Fault |
| s19 | F2 | Fault | s41 | H3 | Fault |
| s20 | F2 | Fault | s42 | Bottom | Boundary |
| s21 | F2 | Fault | | | |

**Table 4. Mapping between interfaces and planar structures.**

| Line | Linear structure | Attribute: type | Line | Linear structure | Attribute: type |
|---|---|---|---|---|---|
| l0 | L1 | Intersection lineation | l26 | L20 | Intersection lineation |
| l1 | L1 | Intersection lineation | l27 | L9 | Intersection lineation |
| l2 | L3 | Intersection lineation | l28 | L18 | Intersection lineation |
| l3 | L1 | Intersection lineation | l29 | L10 | Intersection lineation |
| l4 | L1 | Intersection lineation | l30 | L10 | Intersection lineation |
| l5 | L1 | Intersection lineation | l31 | L11 | Intersection lineation |
| l6 | L2 | Intersection lineation | l32 | L16 | Intersection lineation |
| l7 | L4 | Intersection lineation | l33 | L12 | Intersection lineation |
| l8 | L4 | Intersection lineation | l34 | L15 | Intersection lineation |
| l9 | L3 | Intersection lineation | l35 | L12 | Intersection lineation |
| l10 | L3 | Intersection lineation | l36 | L15 | Intersection lineation |
| l11 | L5 | Intersection lineation | l37 | L16 | Intersection lineation |
| l12 | L6 | Intersection lineation | l38 | L13 | Intersection lineation |
| l13 | L27 | Intersection lineation | l39 | L31 | Intersection lineation |
| l14 | L19 | Intersection lineation | l40 | L14 | Intersection lineation |





| l15 | L20 | Intersection lineation | l41 | L17 | Intersection lineation |
|-----|-----|------------------------|-----|-----|------------------------|
| l16 | L6 | Intersection lineation | l42 | L13 | Intersection lineation |
| l17 | L19 | Intersection lineation | l43 | L15 | Intersection lineation |
| l18 | L19 | Intersection lineation | l44 | L16 | Intersection lineation |
| l19 | L8 | Intersection lineation | l45 | L23 | Intersection lineation |
| l20 | L21 | Intersection lineation | l46 | L22 | Intersection lineation |
| l21 | L29 | Intersection lineation | l47 | L24 | Intersection lineation |
| l22 | L29 | Intersection lineation | l48 | L30 | Intersection lineation |
| l23 | L26 | Intersection lineation | l49 | L22 | Intersection lineation |
| l24 | L25 | Intersection lineation | l50 | L7 | Intersection lineation |
| l25 | L28 | Intersection lineation | l51 | L23 | Intersection lineation |

**Table 5. Mapping between lines and linear structures.**

**4.2 Reconstruction of structural model with semantic description**

The essence of semantic description is spatial geometric data with spatial topological information (basic semantic description) and geological structural meaning (advanced semantic description). Therefore, semantic description can also be used as a source of information for geological modelling. We proposed a method to reconstruct the structural geological model based on semantic description, which proves that the semantic description contains all the structural model information, and is a complete computer characterization of the structural geological model. The steps of the reconstruction algorithm are as follows:

**Input:** *Semantic description of the structural geological model*

**Output:** *Three-dimensional structural geological model*

1. According to the mapping between interfaces and planar structures described in association relations between the basic semantic description and the advanced semantic description, determine whether there are two interfaces with the semantic relation *meet* corresponding to a same planar structure, if there are, go to step 2, if not, go to step 3.

2. Merge two *meet* interfaces into one semantic entity. Delete the line entity corresponding to the overlapping boundary of two interfaces. Delete semantic relations associated with this line entity. Other semantic relations between the two interfaces are retained to the new interface entity. Go to step 1.

3. Choose a 3D surface reconstruction algorithm (Kriging method was used in this paper) to reconstruct all interfaces. Go to step 4.

4. Extract adjacency relations among the reconstructed interfaces by the method of getting basic semantic description mentioned in section 4.1.2 and go to step 5.

5. To judge whether the adjacency relations of interfaces extracted from step 4 is consistent with the semantic description after merging interfaces (by comparing the set of semantic units), if it is, go to step 6, if not, artificially add control points to interface reconstruction process and return to step 3.

6. Close geological bodies according to the association relations between interfaces and bodies. According to the association relations between bodies and stratified structures/massive structures, the attributes geological time can be



given to geological bodies. According to the association relations between interfaces and planar structures, the attributes structure type can be given to the spatial surfaces.

According to the above method and the semantic description of the structural geological model in Xinjiang China obtained in section 4.1, we reconstructed the three-dimensional structural geological model (see figure 16).

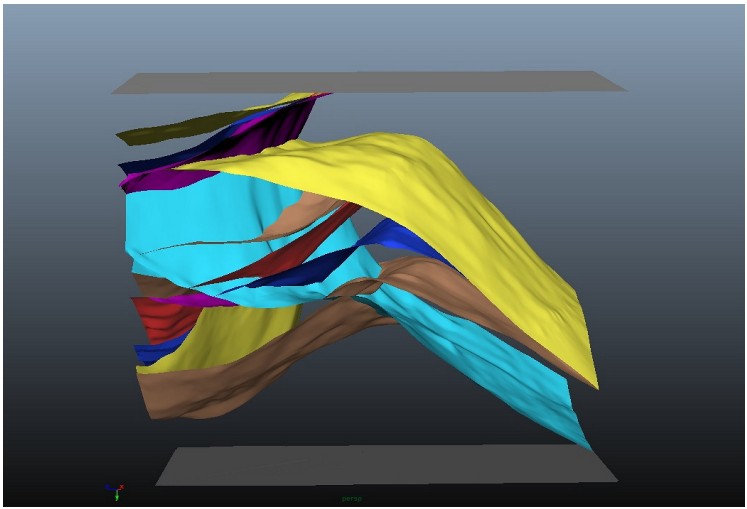

(a) **Stratigraphic framework model.**





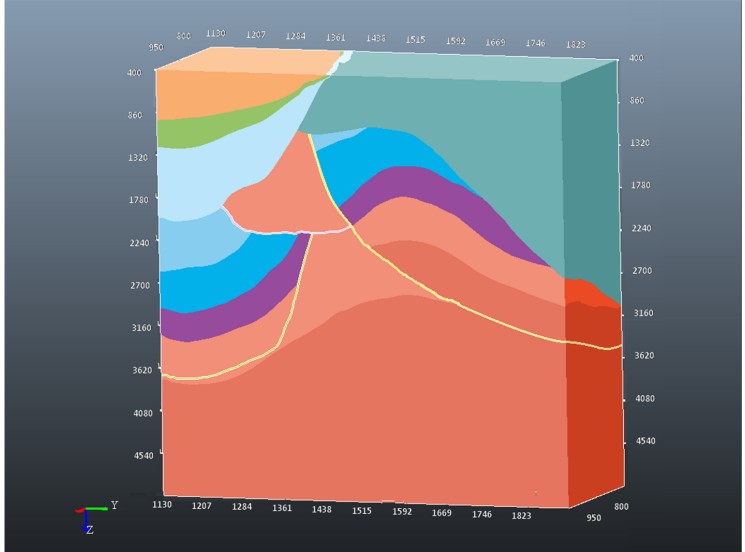

**(b)   3D block model.**

**Figure 16. Three-dimensional structural geological model of a certain area of Xinjiang, China reconstructed according to semantic description. (a)Stratigraphic framework model. (b) Block model. The colour of geological bodies represents the geological time of its formation. Colouring follows the Commission for the Geological Map of the World (Cohen et al., 2013). The figure 16.(a) was generated by Maya v2015, and the figure 16.(b) was generated by the c++ project ComExplore of which the source code was available online and the address is shown in section 8.**

## 5. Structural geological modelling with semantic description

Existing geological modelling methods can be divided into three steps: geological surface reconstruction, geological surface topological relation analysis and three-dimensional solid modelling. Specifically, the first step is reconstructing spatial surfaces with a certain surface reconstruction algorithm according to spatial geometric data from seismic interpretation and drilling. The second step is to determine spatial topological relationships among surfaces (cutting relations of geological surfaces) and generate the stratigraphic framework model. The third step is to take fault planes as internal boundaries of the model to partition geological bodies and generate the solid model.

The reconstruction of structural planes only roughly describes geological structures. We also need to analyse spatial topological relations among structural planes to correctly combine structural planes to get the complete structural model. Existing structural modelling methods reconstruct each surface independently, and then analyse spatial topological relationships of surfaces and combine them into a structural model after reconstructing all surfaces. There are three problems in the topological relations analysis: the rapid calculation of intersection lines between geological surfaces; geological surfaces are not intersected, but the extension of surfaces should be intersected under the constraints of geological structures;



the correct cutting of surfaces when there is a cross between surfaces. Accurate topology is the foundation of the establishment of correct models (Thore et al., 2002). What needs to be emphasized is structural planes are not simply spatial surfaces, but have geological meanings. The form, position and relationships of the surfaces are all restricted by geological laws. Therefore, the topological relations should be under the constraints of geological semantics, which the existing

modelling methods ignore. Structural surfaces are taken as common three-dimensional geometric shapes. This makes it possible to get modelling results which are inconsistent with geological laws when original data have unavoidable uncertainties. The fundamental reason for this problem is that the existing modelling methods are based on computer graphics and image processing methods. In the process of modelling, we lack the restriction of geological semantics and the modelling is driven entirely by original data. At the beginning, the original data can only reflect the local morphological

information of structures, and there is no global structural information of the region, so we call the existing method a bottom-up approach.

We have previously stated that it is feasible to extract semantic description from structural geological model data. And we proved that semantic description contains complete information of geological structures because a structural model can be reconstructed according to the semantic description. Semantic description has determined the structural topology of the

model before finishing modelling, and the modelling process is driven by semantics. So we call the semantics based structural modelling is a kind of top-down modelling method. However, in practical application scenarios, we can't construct the complete semantic description of the structural model without building the model. So semantics-driven top-down modelling needs to be integrated with traditional data-driven bottom-up modelling.

The original data of structural modelling are discrete points in three-dimensional space. These discrete points indicate

structural planes. Geologists can know the rough structure of the structural model through discrete points. So the top-down and bottom-up integrating modelling requires human participation to get rough and incomplete semantic descriptions through human cognition of original data directly. Then a rough structural model can be constructed under the constraints of the rough semantic description. In the initial model, there are some of the most obvious structural planes that have the greatest impact on the model. These structural planes basically determine the nature of the geological structures. Geologists

can use their knowledge and experience of structural geology to guess the state of remaining structural planes and then revise the semantic description. A new structural model can be derived from the new semantic description. This process is repeated until all structures have been extracted with semantic descriptions, modelling is completed, and a complete semantic description is obtained at the same time. In simple terms, this process is a cycle: geologists recognize original data and get semantic description, then implement structural modelling with semantic description as constraints, and details of structures

will be easier to distinguish from original data by geologists with the help of the structural model (see figure 17). Through semantic description, this method integrates geological rules and human experience in structural modelling to solve the problems caused by the lack of geological semantics in traditional methods.





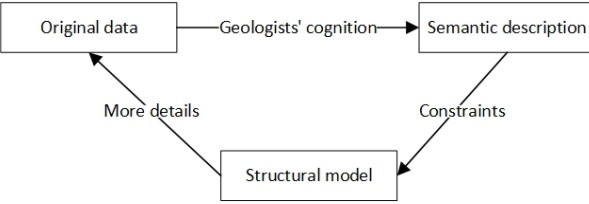

**Figure 17. Structural modelling based on semantic description.**

The core of our proposed bottom-up and top-down integrating modelling approach is structural modelling with the constraints of semantic description. Semantic description constraint structural modelling is to first determine the spatial

topological relationships among structural planes with semantic description, and then perform surface reconstruction. And the topological relationships of structural planes are determined by their boundaries. So what we have to do is to calculate their boundaries based on the semantic description before surface reconstructions, mainly intersection lines of structural surfaces, because intersection lines determine the topological relationships directly.

*Input: Adjacency relations of planar structures, adjacency relations of interfaces, and association relations between*
*planar structures and interfaces.*

*Output: Three-dimensional structural geological model.*

1.    Find planar structures with adjacency relationships *mutual cut, limit and stagger*. According to the mapping between interfaces and planar structures and the adjacency relations of interfaces, find intersected (continuous) interfaces in the same structural plane. As in figure 13(a). $f_1$, $f_2$, $f_3$ are continuous, in figure 13.(b) $f_2$, $f_3$ are continuous, in figure 13

(c). $f_1$, $f_2$ are continuous and $f_5$, $f_6$ are continuous. Merge these continuous interfaces into a new interface entity which is marked with $f_n$ (see figure 18). Go to step 2.

2.    It is assumed that the space of the structure model is a × b (x = 0, 1, … , a; y = 0, 1, … , b). For intersecting interfaces from different planar structures (such as $f_n$ and $f_4$, $f_n$ and $f_5$ in figure 18(a)), interpolate them with n (n ≥ 2) times grid precision in the whole space (x′ = 0, 1/n, 2/n, … , a; y′ = 0, 1/n, 2/n, … , b). The interpolate results are $z_1$

(na × nb) and $z_2$ (na × nb). Calculate absolute values d (na × nb) of the difference between two interpolation results at (x, y): $d_{(x',y')} = \left| z_{1(x',y')} - z_{2(x',y')} \right|$. Go to step 3.

3.    According to the strike of two interpolated interfaces, fix x coordinates or y coordinates as integers . Find the point with minimum absolute value on each grid line parallel to y axis or x axis as the point on the intersecting line of two interfaces:

**for** x=0:a,

**for** y'=0:1/n:b

find (x,y') with min $d_{(x,y')}$

or **for** y=0:b





**for** x'=0:1/n:a

find (x',y) with min $d_{(x',y)}$

And figure 19 shows an example of calculating points on intersection lines. Go to step 4.

4.  Take points on intersection lines as boundaries of interfaces and reconstruct interfaces together with original data. Go to

5       step 5.

5.  Close geological bodies and establish the solid model.

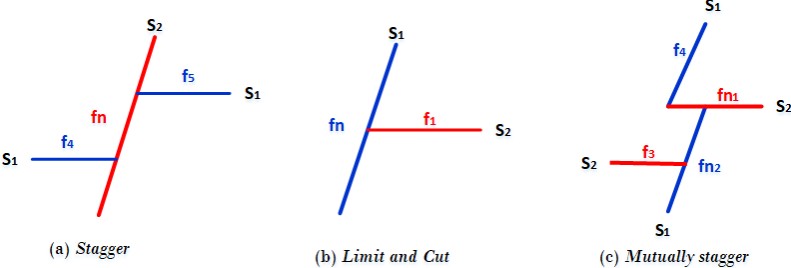

Figure 18. When planar structures have adjacency relations stagger, limit and mutual cut, the relationships among interfaces after interfaces being merged. The merged interfaces were marked with $f_n$.

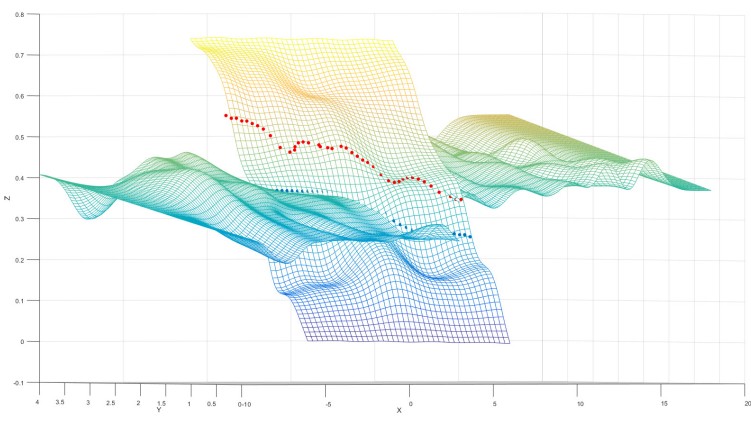

**(a)**



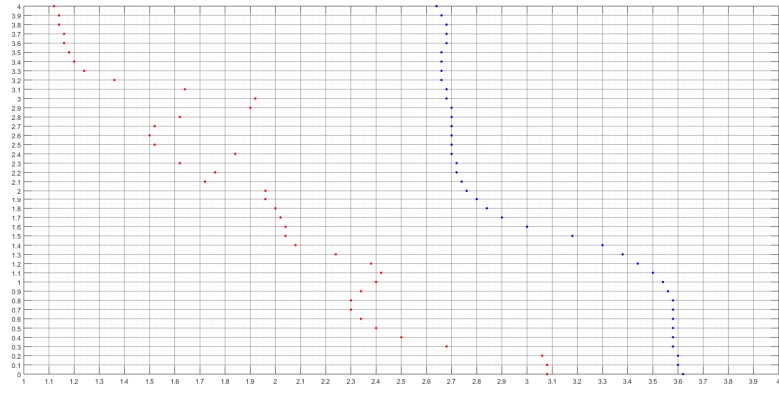

**(b)**

**Figure 19. (a) An example of calculating points on intersection lines. These two planar structures have adjacency relation stagger. There are two intersection lines each with red points and blue points. (b) The projection on the xoy plane of points on intersection**
5   **lines.** `This figure was generated by Matlab v2016a.`

In accordance with the above method, we have realized the 3D structural modelling of an area located Chongqing, China (see figure 20). The transformation of semantic description into boundary information of geological surfaces successfully controls the topological relations of geological surfaces. The top-down and bottom-up integrating modelling process has high reliability because of the addition of semantics. There will be no case where the modelling results don't conform to
10  geological rules due to the data uncertainty and the lack of constraints.



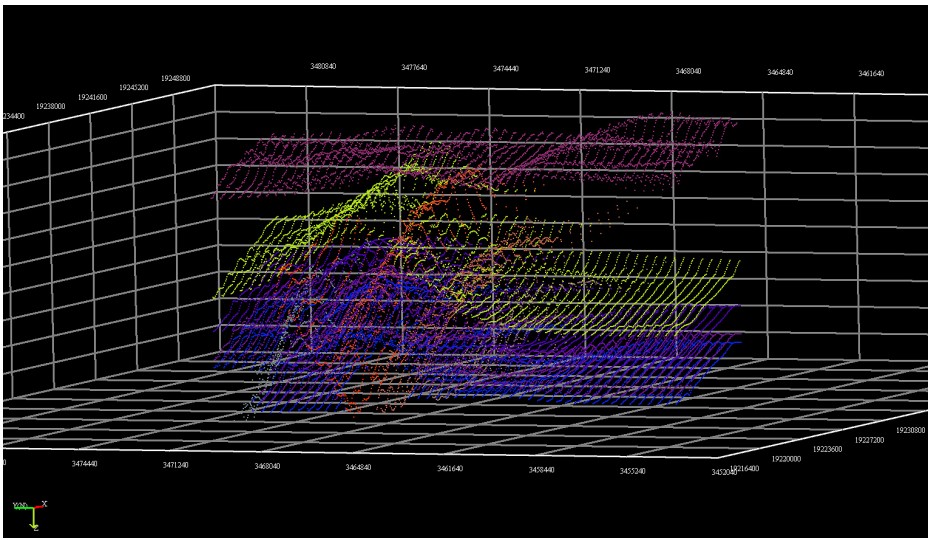

**(a)Original data of the structural model.**

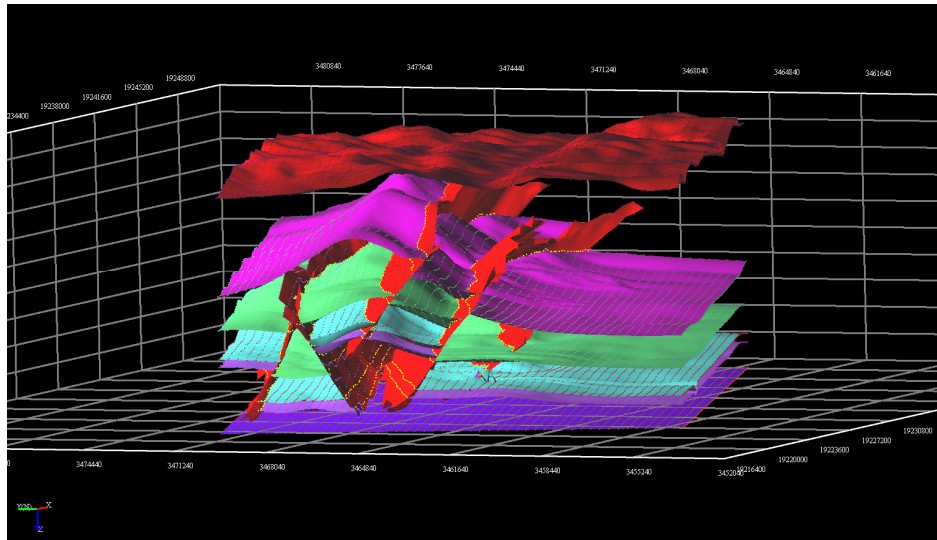

**(b) The stratigraphic framework model.**





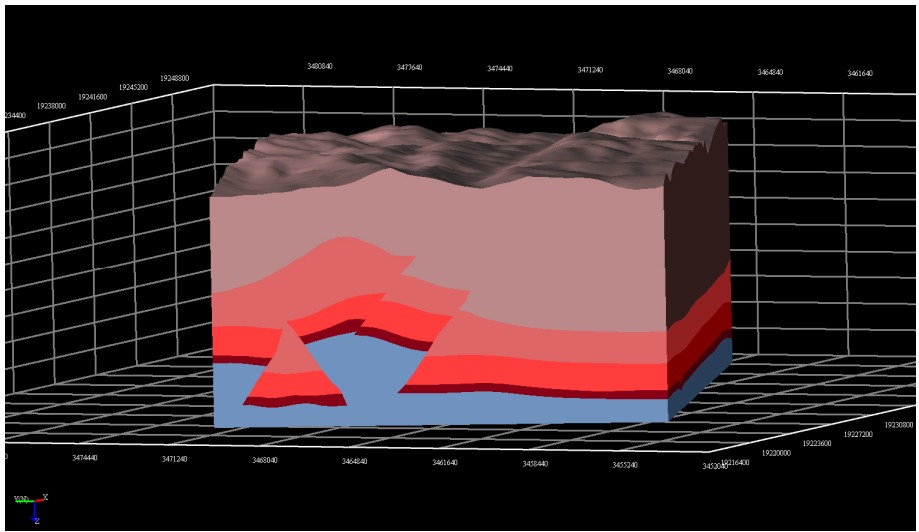

**(c)   The structural geological model.**

**Figure 20. (a) Original data of the structural geological model are discrete points in three-dimensional space. We can roughly see that there are 5 horizons and 4 reverse faults. (b) The stratigraphic framework model after geological surface reconstructions**
**where yellow points are calculated points on intersection lines. (c) The solid structural geological model. This figure was generated by the c++ project ComExplore.**

## 6. Discussions

The semantics of geological elements is the basis for the development and application of artificial intelligence in geosciences in the future, and it is also a key issue for the integration of information science and earth science. Among all geological
elements, structural geological models are key for representing geological structures and geological phenomenon. Therefore, they are the most basic geological elements. So as a first step, we have proposed semantic description of structural geological models. What we need to do in future research is to make good use of the semantic description. In the long run we plan to form semantic representations of all geological elements.

## 7. Conclusions

In conventional geological data models, a lack of semantic description causes difficulty in capturing the full meaning of the data. This often leads to insufficient or inaccurate description of geological structures, especially in cases where geological data is sparse and ambiguous. Existing geological structure modelling methods often suffer instability when they need





artificial adjustment to add control points or control lines for complex structures. There may also be models that do not conform to geological principles.

In this paper, we propose the concept of semantic description of structural geological models. The semantic description of structural geological model is a set of semantic entities, semantic relationships, attributes of semantic entities and original
data. They form a complete computer characterization of three-dimensional structural model. In other words, the semantic description contains complete information of the model and is represented in a computer understandable form. Structural models can be reconstructed according to semantic descriptions. We also propose two algorithms for extracting the semantic description from structural model and reconstructing the structural model with its semantic description. In addition, we propose a new structural modelling process for actual application that uses semantic description as constraints. We call it a
top-down and bottom-up integrating modelling method. It is more reliable because of the introduction of additional geological semantic information to ensure models conform to geological principles.

## 8 Code and data availability

The code for semantic description extraction (mentioned in section 4.1), structural modeling with semantic description (mentioned in section 4.2), and calculating points on intersection lines (mentioned in section 5) together with datasets of the
Xinjiang model and the Chongqing model are available at http://doi.org/10.5281/zenodo.2481084.

## 9 Author contribution

Xianglin Zhan and Guangmin Hu provided ideas. Xianglin Zhan and Jiandong Liang designed the methodology and created models. Xiangling Zhan and Cai Lu did software works. Jiandong Liang did the writing review and editing part.

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
