# Peer review of "Semantic Description and Complete Computer Characterization of Structural Geological Models"

_Geoscientific Model Development, 2018_

## Referee Comment (RC1) · Mark Jessell (Referee) · 12 Apr 2019

This is an interesting paper that attempts to solve an ongoing issue with 3D geological modelling, namely the underutilisation of non-geometric data in model assembly. I agree with the authors about the importance of taking this approach, but have serious reservations about the semantic schema that is presented in the paper as it stands. I encourage the authors to follow the suggestions for improvements in this comment and look forward to further discussion.

A) My first comment is very general which is that the authors fail to acknowledge the considerable efforts that have been made to address this problem in the past, starting

with commercial packages, such as: i. Earthvision: G. Stirewalt and B. Henderson, 1995, A Three-dimensional Geological Framework Model for Yucca Mountain, Nevada, with Hydrologic Application: Report to Accompany 1995 Model Transfer to the Nuclear Regulatory Commission; ii. Geomodeller: Calcagno, P., Chilès, J.P., Courrioux, G., Guillen, A., Geological modelling from field data and geological knowledge, Part I – Modelling method coupling 3D potential-field interpolation and geological rules, Physics of the Earth and Planetary Interiors (2007), doi:10.1016/j.pepi.2008.06.013 iii. Leapfrog: Cowan et al. 2003, Practical implicit modelling, 5th International Mining Conference, AUSIMM. and more recently iv. GemPy: de la Varga et al., 2018, Geosci. Model Dev. Discuss., https://www.geosci-model-dev.net/12/1/2019/gmd-12-1-2019.html In each of the above systems and indeed most modern implicit modelling codes, some semantic knowledge (fault/stratigraphy relationships, fault-fault relationships, stratigraphic column etc.) is a required input for model construction. Even before that, the Noddy system that I developed (sorry for the self-citation! Jessell, M.W. 1981. An interactive Map Creation Package, Unpublished MSc thesis, Unversity of London; Jessell, 1981, Jessell, M.W., Valenta, R.K., 1996. Structural geophysics: integrated structural and geophysical modelling. In: Declan, G.D.P. (Ed.), Computer Methods in the Geosciences. Pergamon, pp. 303e324) transforms a geological history to a 3D model and requires a semantic description of the geology. The authors' reproduction of Figure 1 from Thiele et al. 2016 is described as showing "only a skeleton of structural models, with little additional structural information". Of course this is true, but the authors ignore figures 6,7,8 & 9 from the same article that direclty refer to the semantic nature of the interfaces.

B) My second major concern for a paper that wants to develop a general framework for geology is that it provides a rather poor description of geology in semantic terms, with some important geological characteristics completely ignored, some semantic ideas that are distinct being grouped together, and others that are hierarchical subsets of each other being placed at the same level.

i) As an example of the incompleteness of the semantic description, I include any attempt to have variable internal properties (grain size, chemistry, intensity of foliation, petrophysics, mineralogy, porosity etc.) within volumes, of variable properties (fault thickness, gold grade...) within surfaces. This also means that considerations of hydrothermal overprinting of lithology and metamorphism are ignored. Since most 3D models are actually built to predict these properties, rather than the bulk lithology or structures, this needs to be acknowledged, but is clearly a challenge for the future.

A second aspect that deserves more consideration is the element of time, which here is treated as a simple rock formation-> tectonic event sequence, when in complexly deformed zones there may be multiple episodes of deformation, so that, as with metamorphism, the concept of volumetric overprinting relationships, which are carefully measured in the field, needs to be a part of the analysis if as the authors state, this is to be a complete semantic schema.

The current schema in Fig 8 also ignores all of the structures that may be found in rocks that require a tensor to describe the (crystallographic preferred orientations, anisotropy of magnetic susceptibility or of seismic anisotropy etc.). Although these may be small-scale properties for these models, they may provide important inputs to the modelling process.

ii) There is considerable confusion in the use of the different geological terms (Fig 7). First of all there is the use of non-existent (I believe) terms such as magma squirting and magma condensation, the inclusion of processes that are certainly outside the scope of a schema that has tectonic process as it top level member, such as bioturbation and arguably the input of extra-terrestrial material. Then there is the use of terms such as cementation and crystallisation, where cementation is a form of crystallisation (perhaps the authors meant to distinguish between crystallisation from a magma and from water?).

The terminology related to deformation is also confused. Some terms refer to stress

(extension/compression) some to strain (stretching), and some to volume strain (compaction). It would be wise to stick to strain-based terminology here I think. (Shortening & extension). I suggest the authors look at Means, W. 1976 Stress & Strain Springer. Compression/Stretching combines stress and strain terms in one box.

Secondly, faulting and folding are both processes that can occur when rock masses are extended or compressed, and are hence logically subsets of those behaviours. As previously mentioned, bioturbation is irrelevant at the scale of modelled described in this paper. It is unclear why the inclusion of magma intrusion as a distinct class, which "affects the existing rock masses" is contrasted with crystallisation which can take place in veins, which does not affect them? There could be a scale issue here, but there is nothing inherently different in terms of this discussion in terms of what the process does to the wall rock in veins vs dykes.

In the text describing Fig 8 (page 11, line 13) the authors state that "It should be noted that in structural geology, there is no corresponding association relation concepts between two disjoint structural elements." This worries me as almost every paper on structural geology describing a field study discusses overprinting relationships and relative or absolute timing, and disjoint structural elements are systematically compared. The age difference between two disjoint plutons is the simplest example, and again reflects how much the concept of time needs to be a major part of the analysis.

On page 12 starting at line 24 a series of different semantic definitions related to faults are presented. Fault contact, stagger relation, limit relation, cut, mutually stagger, trace. A fault contact is a clearly defined geometrical and semantic feature, whereas the others do not refer to process: "A limit relation means when a planar structure grows to another planar structure, the younger surface is terminated by the older surface and the younger one does not pass through the older one." These cannot be at the same semantic level, as one includes a notion of process and the other doesn't.

In Figure 8 I am not sure why Stratified structures are classed with Massive structures, to me these seem quite distinct topological concepts. Conversely I am not sure what the distinction is between stratified structures and planar structures is. I believe stratified structures are a sub-class of planar structures, along with penetrative foliations, linear features distributed on a plane etc. I refer the authors to Hobbs, Means & Williams, 1976, An Outline of Structural geology for a more useful description of these concepts.

In order to resolve these problems I urge the authors to collaborate with a field structural geologist so that they can clarify their schema. In addition, I suggest they base their semantic schema on existing ontologies that do not have these problems. Some suggestions:

I) Boyan Brodaric, 2004, The design of GSC FieldLog: ontology-based software for computer aided geological field mapping, Computers & Geosciences 30, 5–20. II) Babaie et al., 2006. Designing a modular architecture for the structural geology ontology. Geological Society of America Special Paper 397. III) Zhong et al., 2009. J. Zhong, A. Aydina, D.L. McGuinness Ontology of fractures J. Struct. Geol., 31 (3) (2009), pp. 251-259 IV) And the intriguing Zhong et al 2006, The ontology of Structural geology, EOS, AGU Abstract. I do not find the full paper for this, but it may be a mine of useful ideas for the current paper.

C) Section 4.2 and section 5 are poorly presented, with insufficient explanation of where the paper is heading. The list of steps in section 4.1.2, 4.2 and on page 32 are extremely hard to follow in their current form, a flow chart with examples of the steps would be much more useful.

D) The discussion, once the paper acknowledges properly previous work in the field, should show which parts of the work are new, and how they have advanced the field. It is much too short at present, and simply asserting that the new method is more reliable without direct testing against other methods is of no value.

Other minor comments

a) Figure 5 needs to be properly cited, even if though the author allows open reproduction b) Line 10: "Semantic description is the interpretation of an object at the semantic level". Circular definitions are not very useful! c) The caption for Figure 15 refers to features that are impossible to see in the figures, either through the choice of yellow text, or through the lack of labels (I am not sure). For example: "(c) it can be seen that there are 25 points on the line l40". I do not see a label l40, and I cannot see which 25 points are referred to? Most of the observations in the caption are not visible in the associated figure. d) Figure 18 and Figure 13 overlap so much that only Figure 13 is needed.

---

## Referee Comment (RC2) · Florian Wellmann (Referee) · 23 Apr 2019

In the manuscript with the title "Semantic Description and Complete Computer Characterization of Structural Geological Models", the authors present an interesting approach for a semantic description of geological elements that are commonly used in the context of structural geological modelling. The approach is fitting very well into the evolving topic of topological and semantic analyses of geological models.

In this context, however, the exact contribution of the work is not entirely clear to me. The authors combine a very detailed semantic description of geological models with an application study and my further comments related to these two points separately.

Concerning the first point, many aspects of the included semantic description show a lot of similarity with the description in Thiele et al., 2016. Even though this paper is briefly referenced in the introduction, this similarity is not evident in the following own contribution in section 2. It is correct that the work of Thiele et al. focussed on the topological analysis, but it also went beyond a pure description of topological relationships and included geological aspects. In the same way as this manuscript, the work in Thiele and al was motivated by the Egenhofer and Hering (1990) paper, and the semantic associations in Fig. 3 of this manuscript are identical to the ones described in Fig. 1 of Thiele et al. To be sure, the more detailed analysis of the 9-intersection model provided here adds interesting aspects, but the relevance of these aspects is not entirely clear (note that Thiele et al. also describe temporal relationships - so, in fact, what is implemented here with the definition of primary and secondary structures on page 13, lines 8 ff.). In the terminology of the authors, the description of Thiele and al. also includes "advanced semantic entities". Also, the authors only describe "primary" and "secondary" elements, but many geological systems are clearly affected by more than two tectonic events.

The semantic description is then applied in a case study to evaluate how adding this information improves model construction. In this application study, it seems that the authors are applying the concepts mainly to overcome problems in the specific interpolation approach they are using. However, there are by now many modeling approaches that include aspects of geological reasoning (e.g. Calcagno et al., 2009; Mallet et al., 2004; see also our recent overview in Wellmann and Caumon, 2018 for more references), as well as "advanced semantic aspects" like unconformities, faults, intrusions, etc. This does not mean that the analysis of topology may not add very important aspects that these methods still do not consider, but it should be mentioned more clearly what exactly the authors aim to add.

In my understanding, the main contribution in the case study is that the authors use semantic entities to quickly evaluate if a generated geological interpolation conforms to

the expected setting. If my interpretation is correct, then it would be good to focus the case study on this aspect and to describe more clearly how the semantic relations are estimated from (independent?) data. In Part 3./ page 21, the authors only describe that this information is taken from seismic data - but if this is the case, then is this based on 2-D or 3-D seismic data? And if 2-D: how many lines, and how is it evaluated if the 2-D analysis is really representative of the 3-D topology? In the section on model reconstruction (4.2), the semantic description is then used as a way to check model modifications - but it is not clear on which basis, for example, control points are added (line 22). Maybe a simple example would help here.

In section 5, the authors then describe their iterative approach of semantic evaluation and model construction. As stated before, this is a very interesting aspect in this paper. However, in the motivation of the approach, it is simply stated that existing modeling methods ignore these semantic aspects (page 31, line 5) - a statement that is (1) given without any references, and (2) not generally correct (see comments above). A clearer description of the own contribution would be helpful here, and a more detailed comparison to existing literature.

The organisation of the manuscript is overall clear, with the definition of the methods and the application in a case study. One aspect that should be adapted is the mixture between the "Methods" section 4 with the actual case study. It would be better to clearly separate both parts, or completely combine them into a section "Case study". I personally found the detailed "workflow" descriptions in section 4.1.2 more confusing than valuable. Maybe a graphical representation in a workflow diagram would be better suited here.

Overall, the manuscript is written in clearly understandable scientific English. Some parts would benefit from a more thorough proof-reading, with several (minor) grammatical mistakes and unclear sentences. Some of the terminology in the section on the semantic elements is not consistent with commonly used terms in the field of structural geology - a thorough checking of these terms would be helpful. The figures are

generally clear and helpful, but some information is a bit redundant (e.g. Fig. 17) and several figures could potentially be combined in fewer figures (the manuscript currently contains 20 figures).

In summary, the manuscript contains many interesting aspects - but lacks almost completely references to previous work and other modeling approaches. This aspect is especially evident in the (very short) discussion, which does not place the contribution into the context of existing literature. Without a more detailed placement of the own work in the context of this previous work, the scientific contribution can hardly be judged. This refers to both the semantic description, as well as to the application in the case study. With more clarity about this aspect, the work could potentially add very interesting aspects to the field.

―――――――――――――――――――

---

## Author Comment (AC1) · 1 May 2019

**Xianglin Zhan et al.**

xianglin\_zhan@163.com

Received and published: 1 May 2019

Dear Prof. Jessell,

Thank you for your comprehensive, detailed, and constructive comments. Here we would like to address your main concerns and outline how we plan to address all the points. The detailed changes to the manuscript will be included in the next revised version of the manuscript.

First, let me address what we understand to be your primary concerns, in point B. : "i). As an example of the incompleteness of the semantic description, I include any

attempt to have variable internal properties (grain size, chemistry, intensity of foliation, petrophysics, mineralogy, porosity etc.) within volumes, of variable properties (fault thickness, gold grade...) within surfaces. This also means that considerations of hydrothermal overprinting of lithology and metamorphism are ignored. Since most 3D models are actually built to predict these properties, rather than the bulk lithology or structures, this needs to be acknowledged, but is clearly a challenge for the future. A second aspect that deserves more consideration is the element of time, which here is treated as a simple rock formation-> tectonic event sequence, when in complexly deformed zones there may be multiple episodes of deformation, so that, as with metamorphism, the concept of volumetric overprinting relationships, which are carefully measured in the field, needs to be a part of the analysis if as the authors state, this is to be a complete semantic schema. The current schema in Fig 8 also ignores all of the structures that may be found in rocks that require a tensor to describe the (crystallographic preferred orientations, anisotropy of magnetic susceptibility or of seismic anisotropy etc.). Although these may be small-scale properties for these models, they may provide important inputs to the modelling process."

Response: We truly appreciate you spotting the lack of clear description on the scope for our semantic description. Although we have a wide and long-term perspective in mind, we intend to limit the effort with this manuscript to a, hopefully, manageable scope. The improtance of such a restriction becomes more evident now your comments have revealed the inadequacy of our knowledge in geology. We will make it clear in our reversion that we intend only to improve the semantic aspect of structural geological models as a first step. The hope is, such an effort would eventually lead to an ever widening scope, such as petrophysical and chemical properties, mineral properties, microscale structures, crystal properties, etc., as you pointed out. We plan to collaborate with experts in respective domains so as to gradually expand the scope and improve the completeness of semantic descriptions.

The following are responses to other comments of yours. "A) My first comment is very

**GMDD**
general which is that the authors fail to acknowledge the considerable efforts that have been made to address this problem in the past, starting with commercial packages, such as: i. Earthvision: G. Stirewalt and B. Henderson, 1995, A Three-dimensional Geological Framework Model for Yucca Mountain, Nevada, with Hydrologic Application: Report to Accompany 1995 Model Transfer to the Nuclear Regulatory Commission; ii. Geomodeller: Calcagno, P., Chilès, J.P., Courrioux, G., Guillen, A., Geological modelling from field data and geological knowledge, Part I - Modelling method coupling 3D potential-field interpolation and geological rules, Physics of the Earth and Planetary Interiors (2007), doi:10.1016/j.pepi.2008.06.013 iii. Leapfrog: Cowan et al. 2003, Practical implicit modelling, 5th International Mining Conference, AUSIMM. and more recently iv. GemPy: de la Varga et al., 2018, Geosci. Model Dev. Discuss., https://www.geosci-model-dev.net/12/1/2019/gmd-12-1-2019.html In each of the above systems and indeed most modern implicit modelling codes, some semantic knowledge (fault/stratigraphy relationships, fault-fault relationships, stratigraphic column etc.) is a required input for model construction. Even before that, the Noddy system that I developed (sorry for the self-citation! Jessell, M.W. 1981. An interactive Map Creation Package, Unpublished MSc thesis, Unversity of London; Jessell, 1981, Jessell, M.W., Valenta, R.K., 1996. Structural geophysics: integrated structural and geophysical modelling. In: Declan, G.D.P. (Ed.), Computer Methods in the Geosciences. Pergamon, pp. 303e324) transforms a geological history to a 3D model and requires a semantic description of the geology. The authors' reproduction of Figure 1 from Thiele et al. 2016 is described as showing "only a skeleton of structural models, with little additional structural information". Of course this is true, but the authors ignore figures 6,7,8 & 9 from the same article that directly refer to the semantic nature of the interfaces."

Response: Thank you for your detailed list of prior arts. We will study them thoroughly and acknowledge their contributions. We acknowledge that we misunderstood the topology system proposed in Thiele et al. 2016 as a mere description of the topological relationships of geological bodies, ignoring the implicit semantic nature. We will correct our conclusion on this.
"B) ii) There is considerable confusion in the use of the different geological terms (Fig 7). First of all there is the use of non-existent (I believe) terms such as magma squirting and magma condensation, the inclusion of processes that are certainly outside the scope of a schema that has tectonic process as it top level member, such as bioturbation and arguably the input of extra-terrestrial material. Then there is the use of terms such as cementation and crystallisation, where cementation is a form of crystallization (perhaps the authors meant to distinguish between crystallisation from a magma and from water?). The terminology related to deformation is also confused. Some terms refer to stress" (extension/compression) some to strain (stretching), and some to volume strain (compaction). It would be wise to stick to strain-based terminology here I think. (Shortening & extension). I suggest the authors look at Means, W. 1976 Stress & Strain Springer. Compression/Stretching combines stress and strain terms in one box. Secondly, faulting and folding are both processes that can occur when rock masses are extended or compressed, and are hence logically subsets of those behaviours. As previously mentioned, bioturbation is irrelevant at the scale of modelled described in this paper. It is unclear why the inclusion of magma intrusion as a distinct class, which "affects the existing rock masses" is contrasted with crystallisation which can take place in veins, which does not affect them? There could be a scale issue here, but there is nothing inherently different in terms of this discussion in terms of what the process does to the wall rock in veins vs dykes. In the text describing Fig 8 (page 11, line 13) the authors state that "It should be noted that in structural geology, there is no corresponding association relation concepts between two disjoint structural elements." This worries me as almost every paper on structural geology describing a field study discusses overprinting relationships and relative or absolute timing, and disjoint structural elements are systematically compared. The age difference between two disjoint plutons is the simplest example, and again reflects how much the concept of time needs to be a major part of the analysis. On page 12 starting at line 24 a series of different sematic definitions related to faults are presented. Fault contact, stagger relation, limit relation, cut, mutually stagger, trace. A fault contact is a clearly defined

**GMDD**
geometrical and semantic feature, whereas the others do not refer to process: "A limit relation means when a planar structure grows to another planar structure, the younger surface is terminated by the older surface and the younger one does not pass through the older one." These cannot be at the same semantic level, as one includes a notion of process and the other doesn't. In Figure 8 I am not sure why Stratified structures are classed with Massive structures, to me these seem quite distinct topological concepts. Conversely I am not sure what the distinction is between stratified structures and planar structures is. I believe stratified structures are a sub-class of planar structures, along with penetrative foliations, linear features distributed on a plane etc. I refer the authors to Hobbs, Means & Williams, 1976, An Outline of Structural geology for a more useful description of these concepts. In order to resolve these problems I urge the authors to collaborate with a field structural geologist so that they can clarify their schema. In addition, I suggest they base their semantic schema on existing ontologies that do not have these problems."

Response: As mentioned above, thanks to your comprehensive and detailed comments, we now recognize our lack of knowledge in geology more than before. We plan to correct the confusions of concepts and misuse of terms as you mentioned. Furthermore, we have just started looking for experts in geology to help us address other issues you indicated, and to continue this effort on an on-going basis. Since you are a well-known expert in this field, we would very much appreciate any opportunity to work with you on this.

"C) Section 4.2 and section 5 are poorly presented, with insufficient explanation of where the paper is heading. The list of steps in section 4.1.2, 4.2 and on page 32 are extremely hard to follow in their current form, a flow chart with examples of the steps would be much more useful."

Response: The purpose of Section 4.2 is to illustrate the completeness of semantic descriptions, within our limited scope, by reconstructing models from semantic descriptions. The main content of Section 5 is to illustrate the role that semantic description
can play in the practical structural modeling process. The content of these two sections has less relevance to the semantic description itself. So we will revise to make it clearer and more succinct, following your suggestion of using flowcharts

"D) The discussion, once the paper acknowledges properly previous work in the field, should show which parts of the work are new, and how they have advanced the field. It is much too short at present, and simply asserting that the new method is more reliable without direct testing against other methods is of no value."

Response: We will clarify our motivation and state what we consider to be our contribution within our priscribed scope. We plan to add simulation experiments to illustrate the limitations of existing structural modeling methods under certain situation and compare them with the results of our methods.

"a) Figure 5 needs to be properly cited, even if though the author allows open reproduction."

Response: Than you for pointing out this. We will add a reference.

"b) Line 10: "Semantic description is the interpretation of an object at the semantic level". Circular definitions are not very useful! "

Response: We will refine the definition to avoid circular definition.

"c) The caption for Figure 15 refers to features that are impossible to see in the figures, either through the choice of yellow text, or through the lack of labels (I am not sure). For example: "(c) it can be seen that there are 25 points on the line I40". I do not see a label I40, and I cannot see which 25 points are referred to? Most of the observations in the caption are not visible in the associated figure."

Response: Due to the size of the network, we did not display all the labels of nodes and edges. Figure 15 was originally an "interactive image", and each node and edge would display information once highlighted, such as shown in subfigure (c), (d) and (e) of figure 15. We will correct the caption of Figure 15 and try to improve the presentation

GMDD
of Figure 15. We will also provide the interactive source files for Figure 15. "d) Figure 18 and Figure 13 overlap so much that only Figure 13 is needed." Response: Figure 18 will be removed.

---

## Author Comment (AC2) · 8 May 2019

Dear Prof. Wellmann,

On behalf of all the co-authors, I would like to express our sincere appreciation for your comprehensive, detailed, and constructive comments. I will first address main concerns you have and outline how we plan to address all the points. The detailed changes will be included in the revision. To facilitate discussion, I labelled your comments with numbered points.

1. Your comment: "Concerning the first point, many aspects of the included semantic

description show a lot of similarity with the description in Thiele et al., 2016. Even though this paper is briefly referenced in the introduction, this similarity is not evident in the following own contribution in section 2. It is correct that the work of Thiele et al. focused on the topological analysis, but it also went beyond a pure description of topological relationships and included geological aspects. In the same way as this manuscript, the work in Thiele and al was motivated by the Egenhofer and Hering (1990) paper, and the semantic associations in Fig. 3 of this manuscript are identical to the ones described in Fig. 1 of Thiele et al. To be sure, the more detailed analysis of the 9-intersection model provided here adds interesting aspects, but the relevance of these aspects is not entirely clear (note that Thiele et al. also describe temporal relationships - so, in fact, what is implemented here with the definition of primary and secondary structures on page 13, lines 8 ff.). In the terminology of the authors, the description of Thiele and al. also includes "advanced semantic entities"."

Response: Thank you for your comprehensive and thorough comments. First of all, we highly value the contributions made by Thiele et al., and we strive to push forward with our own work on that basis. We had misunderstood the topology system proposed in Thiele et al. as a mere description of the topological relationships of geological bodies, ignoring the implicit semantic nature. We will correct our conclusion on this. We now realized the structural topology proposed by Thiele et al. can also semantically describe the construction model to a certain extent. However, we would argue that we still improved the completeness of description. For example, we explicitly added description of geometric shapes, included relationships between geometric elements of different dimensions, and our structural model is no longer segmented from a pure geometric perspective. We will definitely clarify the contribution made by Thiele et al. in our revision, and clearly specify both the relevance and differences between our work and Thiele's work. As for the similarity between Fig. 1 of Thiele et al. and Fig. 3 of our manuscript, in our understanding, Fig. 1 of Thiele et al. emphasizes relative positional relations of two geometric elements of the same dimension, while Fig. 3 of ours emphasize the subordinate relationships between geometric elements

from different dimensions. So we think these two figures differ in meanings and we will revise to make this clear.

2.Your comment: "Also, the authors only describe "primary" and "secondary" elements, but many geological systems are clearly affected by more than two tectonic events."

Response: "Primary" and "secondary" are derived from the concepts "primary structure" and "secondary structure". The primary tectonic events refer to all tectonic events that directly affect the existence of rock masses, while the secondary tectonic events refer to tectonic events that only deform rock masses. So the fact that elements are divided into primary and secondary does not imply that only two tectonic events have affected the geological system. We will revise to avoid this possible confusion.

3.Your comment: "The semantic description is then applied in a case study to evaluate how adding this information improves model construction. In this application study, it seems that the authors are applying the concepts mainly to overcome problems in the specific interpolation approach they are using. However, there are by now many modeling approaches that include aspects of geological reasoning (e.g. Calcagno et al., 2009; Mallet et al., 2004; see also our recent overview in Wellmann and Caumon, 2018 for more references), as well as "advanced semantic aspects" like unconformities, faults, intrusions, etc. This does not mean that the analysis of topology may not add very important aspects that these methods still do not consider, but it should be mentioned more clearly what exactly the authors aim to add."

Response: We truly appreciate such a detailed list of prior arts. We will study them thoroughly and acknowledge their contributions. We will clarify the improvement of our semantic description comparing to the topological analysis in the revision.

4. Your comment:"In my understanding, the main contribution in the case study is that the authors use semantic entities to quickly evaluate if a generated geological interpolation conforms to the expected setting. If my interpretation is correct, then it would be good to focus the case study on this aspect and to describe more clearly how

the semantic relations are estimated from (independent?) data."

Response: Our semantic description does have the function of evaluating whether the model is consistent with settings. However, in this article we explain the practical application value of semantic description from the perspective of constraint modeling process. We hope that the structural modeling with semantic description constraints can directly achieve the certain expected result instead of evaluating after modeling is complete. The estimation of the semantic description is basically based on the expert's cognition of the raw data and the computer's estimation of the geometric topological relationships between two interfaces. We will take your suggestion to describe the potential of our semantic description on model check and give a more detailed description on how to estimate the semantic description.

5.Your comment: "In Part 3./ page 21, the authors only describe that this information is taken from seismic data - but if this is the case, then is this based on 2-D or 3-D seismic data? And if 2-D: how many lines, and how is it evaluated if the 2-D analysis is really representative of the 3-D topology?"

Response: Our basic data comes from structural interpretation of 3D seismic data. We will revise to make this clear.

6.Your comment: "In the section on model reconstruction (4.2), the semantic description is then used as a way to check model modifications - but it is not clear on which basis, for example, control points are added (line 22). Maybe a simple example would help here."

Response: Very helpful suggestion. Although the semantic description provides constraints on the wireframes of the structural model, the surface reconstruction algorithm itself is unconstrained. That is to say, the morphological details inside the surface boundaries are not completely constrained. When the surface is reconstructed, there may be cases where the topology and the semantic description becomes inconsistent due to the shape of the surface (e.g. the crossing between stratigraphic planes.), so we

set this inspection step using the semantic description here. Only when such situation occurs does it becomes necessary to artificially add control points to correct the error in surface morphology. The control points are set based on the distance between the incorrect interpolation result and the correct interval specified by the semantic description. We will give some simple illustrations in the revision to show when the control points are needed and how to add them.

7.Your comment: " In section 5, the authors then describe their iterative approach of semantic evaluation and model construction. As stated before, this is a very interesting aspect in this paper. However, in the motivation of the approach, it is simply stated that existing modeling methods ignore these semantic aspects (page 31, line 5) - a statement that is (1) given without any references, and (2) not generally correct (see comments above). A clearer description of the own contribution would be helpful here, and a more detailed comparison to existing literature."

Response: As mentioned above, thanks to your comprehensive and detailed comments, we now recognize our lack of acknowledgement to existing geological semantic description researches. We will re-evaluate the contribution of these methods and add references. We also plan to give more simulation experiments to illustrate the innovation of our semantic description.

8.Your comment:ãĂĂ"The organisation of the manuscript is overall clear, with the definition of the methods and the application in a case study. One aspect that should be adapted is the mixture between the "Methods" section 4 with the actual case study. It would be better to clearly separate both parts, or completely combine them into a section "Case study"."

Response: The purpose of Section 4 is to clarify how to extract our semantic description from structural models (Section 4.1) and prove the completeness of the semantic description (Section 4.2), while Section 5 focuses on how one can use the semantic description to solve practical modeling problems. The former one theoretically illustrates the feasibility of extracting the semantic description and the completeness of the semantic description. The latter one explains the application value of semantic descriptions in practical applications. Therefore we intend to separate the two sections more thoroughly and clarify their focal points.

9.Your comment: "I personally found the detailed "workflow" descriptions in section 4.1.2 more confusing than valuable. Maybe a graphical representation in a workflow diagram would be better suited here."

Response: We will revise to make it clearer and more succinct, following your suggestion of using flowcharts.

10.Your comment: "Some parts would benefit from a more thorough proof-reading, with several (minor) grammatical mistakes and unclear sentences."

Response: We will make a more careful proofreading of the manuscript in the revision.

11.Your comment: "Some of the terminology in the section on the semantic elements is not consistent with commonly used terms in the field of structural geology - a thorough checking of these terms would be helpful."

Response: Thank you for pointing out our inaccuracy in the use of terms. We plan to correct the confusions of concepts and misuse of terms as you mentioned.

12.Your comment: "The figures are generally clear and helpful, but some information is a bit redundant (e.g. Fig. 17) and several figures could potentially be combined in fewer figures (the manuscript currently contains 20 figures)."

Response: Fig. 17 will be removed or replaced by a more useful and detailed figure to illustrate our idea about structural modeling. We also plan to follow your suggestion to combine unnecessary figures (e.g. Fig. 18 and Fig. 13).